# The structured backbone of temporal social ties

Teruyoshi Kobayashi [1], Taro Takaguchi[2] & Alain Barrat [3,4]

In many data sets, information on the structure and temporality of a system coexists with noise and non-essential elements. In networked systems for instance, some edges might be non-essential or exist only by chance. Filtering them out and extracting a set of relevant connections is a non-trivial task. Moreover, mehods put forward until now do not deal with time-resolved network data, which have become increasingly available. Here we develop a method for filtering temporal network data, by defining an adequate temporal null model that allows us to identify pairs of nodes having more interactions than expected given their activities: the significant ties. Moreover, our method can assign a significance to complex structures such as triads of simultaneous interactions, an impossible task for methods based on static representations. Our results hint at ways to represent temporal networks for use in data-driven models.

[1] Graduate School of Economics, Center for Computational Social Science, Kobe University, Kobe, Japan. [2] Toda, Saitama 335-0021, Japan. [3] Aix Marseille Univ, Université de Toulon, CNRS, CPT, Marseille 13009, France. [4] Data Science Laboratory, ISI Foundation, Torino 10126, Italy. Correspondence and requests for materials should be addressed to A.B. (email: Alain.Barrat@cpt.univ-mrs.fr)

The analysis of large-scale empirical data sets, and in particular of complex networked data, is often made difficult by the nature of the data itself: data may be noisy[1–3], and contain both robust, generalizable properties and details specific to the collected data set under investigation, which would change if the data had been collected at a different moment or for a different sample of the same system. For instance, data describing interactions between individuals (face-to-face[4,5], phone calls[6,7], or online interactions[8,9]) might show robust group structures in different days or weeks but with a different set and timing of interactions each day: the exact timing of an interaction in a specific day might not be relevant to the understanding of the population's characteristics. Another issue might arise if the network under scrutiny carries very heterogeneous weights on the edges. The importance of edges might then not be easily reducible only to their own weight, nor to the local properties of the nodes they link, such as their degree (number of neighbors in the network) or their strength (sum of the weights of their edges).

In order to extract the most relevant information from the data, several approaches have been put forward for static networks. For instance, the k-core decomposition focuses on more and more connected parts of a network and has been established as an important tool to analyze and visualize complex networks and to determine influential spreaders in networks[10–12]. Another approach consists in determining a "backbone" of significant edges in the network, and to filter out the remaining nonessential edges. Several methods have been proposed to this purpose in the case of static weighted networks. The simplest way of filtering edges is through thresholding: all the edges with weight below a given threshold value are removed. Such a method, however, imposes an arbitrary cutoff scale, while many systems of interest display broad distributions of weights and complex patterns at multiple scales. Other methods have thus been put forward to filter out edges simultaneously at different scales, using statistical tests based on null models[13–19]: the fundamental idea is to test whether the weight of an edge is distinguishable from the hypothetical one that would be generated at random by a certain null model. Filtering is performed by fixing a desired significance level and selecting only those edges whose weight cannot be explained by the null model at the chosen significance level. These significant edges form a backbone of the network.

Various null models have been proposed in the literature to deal with static weighted networks[13–19]. The recent surge in the availability of temporally resolved high-resolution data on social and economic networks highlights, however, the need for methods specifically designed to extract backbones from temporal networks or temporally aggregated networks[20,21]. Obviously, each method defined on static weighted methods can be applied to a temporally aggregated network: for instance, a simple threshold could be applied on the numbers of events (i.e., on the weights of the aggregated edge) between two nodes[22]. However, a highly active node could in principle have a large number of (nonessential) ties, so that one needs to control for the difference in intrinsic activity levels across nodes to extract statistically significant ties that cannot be explained by random chance.

Here, we develop a method to extract an irreducible backbone from a sequence of temporal contacts between nodes, by defining an adequate temporal null model. This null model can be interpreted as a (temporal) configuration or fitness model, whose parameters are estimated by using global information (the numbers of contacts for all node pairs), similarly to the enhanced configuration model (ECM) filter for static networks[17]. Thanks to this null model, we determine the set of significant ties, at any significance level, among all the pairs of nodes having interacted. These ties form an irreducible backbone in the sense defined in ref. [17], as their significance cannot be reduced to the activity of the involved nodes. Most importantly, the temporal nature of the null model allows us to attribute a significance also to higher-order structures, such as simultaneously occurring triplets of interactions or other temporal motifs[23]. By construction, such a task would be impossible when defining significant ties and backbones directly from a temporally aggregated network.

## Results

**Main outcomes**. In the following, we validate our filtering method on a synthetic benchmark and illustrate its application on temporal networks of social and economic relevance. We compare its results with several static filtering methods and with baseline temporal extensions of static filters, obtained by simply applying the same static filter to the successive snapshots of the temporal network. Interestingly, at a given level of significance, our method generally identifies more significant edges than other filters and performs better at retrieving known significant edges of synthetic benchmarks. We also show that our method is able to detect significant edges at all scales of interaction intensity. Moreover, in cases where the aggregated network has a clear-cut community structure, corresponding, e.g., to classes in a school, the significant ties turn out to be mostly intra-community ones. At high significance levels, the network of significant ties (i.e., backbone) breaks into several connected components, each corresponding to one community, and inter-community edges turn out to be nonsignificant. This suggests that inter-community edges, while playing a crucial role in reducing the diameter of the network[24,25], are here indistinguishable from randomly created edges, once nodes' activities are fixed. We also illustrate the ability of our filtering method to assign a significance to higher-order structures by investigating significant triads, defined as sets of three nodes that interact simultaneously with each other more than expected given their activities. Strikingly, it turns out that these significant triads are not necessarily composed of three significant edges. This shows the crucial importance of taking into account temporality when defining a null model to detect the significance of structures in temporal networks, as such information could not be obtained from a purely static null model nor from the simple extension obtained by applying a static filter to each temporal snapshot.

**Data**. We consider eight data sets describing systems of very different nature and of social and economic interest, described by temporal networks (Table 1). The first four data sets correspond to face-to-face contacts among individuals in different contexts, recorded using wearable sensors by the SocioPatterns collaboration with a temporal resolution of 20 s and publicly available (http://www.sociopatterns.org/datasets). We consider data sets collected in contexts with very different activity levels, constraints on the schedule of individuals, duration and group structures, namely a high school ("Highschool")[26], a primary school ("Primaryschool")[5], an office building ("Workplace")[27], and a hospital ward ("Hospital")[28]. The fifth data set, "Interbank", is a temporal financial network in which nodes and edges represent banks and overnight lending–borrowing relationships, respectively. Since overnight loan contracts last only for 1 day, we can construct a sequence of daily snapshot networks (i.e., time resolution is 1 day)[29,30]. We consider here the data on the online interbank market in Italy, called e-MID, between June 12, 2007 and July 9, 2007 (i.e., 20 business days). The data are commercially available from e-MID SIM S.p.A. based in Milan, Italy (http://www.e-mid.it). The sixth data set is the temporal network of emails exchanged between members of a European research institution ("Email"), downloaded from http://snap.stanford.edu/data/index.html. In the Email data, we consider daily

**Table 1 Basic description of empirical temporal networks**

| Data | N | # Temporal edges | # Aggregate edges | Time span | # Communities | Q |
|---|---|---|---|---|---|---|
| Highschool | 327 | 188,508 | 5818 | 5 days | 9 | 0.809 |
| Primaryschool | 242 | 125,773 | 8317 | 2 days | 10 | 0.627 |
| Workplace | 217 | 78,249 | 4274 | 10 days | 12 | 0.624 |
| Hospital | 75 | 32,424 | 1139 | 5 days | 4 | 0.215 |
| Interbank | 162 | 7104 | 2140 | 20 days | 2 | 0.036 |
| Email | 986 | 332,334 | 16,064 | 526 days | 42 | 0.669 |
| LondonBike | 743 | 38,023 | 18,752 | 24 h | 5 | 0.424 |
| UK-airline | 55 | 2787 | 398 | 14 years | 3 | 0.081 |

For the Interbank data, the number shown in the third column denotes the number of daily edges rather than the total number of transactions. The "# communities" column gives the number of classes for the Primaryschool and Highschool data, of office departments for Workplace, and of types of occupations for the Hospital data. We classify banks into two groups, Italian banks and foreign banks. For the Email, LondonBike, and UK-airline datasets, communities are detected by applying Infomap[44] on the aggregate weighted network. Q is the weighted modularity of the corresponding partition

network snapshots in which nodes represent individuals and an edge between two nodes denotes the presence of at least one email exchanged between the corresponding persons on a given day. The seventh one describes the trips taken by customers of London Bicycle Sharing Scheme ("LondonBike")[31]. The nodes represent bike-sharing stations and edges denote the presence of trips between two stations on June 22, 2014. Finally, the eighth data set, downloaded from http://www.bifi.es/~cardillo/data.html, is given by the UK domestic airline network between 1990 and 2003 ("UK-airline") [32], at yearly resolution, in which nodes and edges denote the UK airports and the presence of direct flights connecting two airports. For all data sets, we consider undirected networks. More details about data processing are provided in the section "Methods".

**Temporal fitness model.** We consider a set of $N$ nodes and a sequence of interactions that occur at arbitrary points in time between these nodes[21,33]. We fix a temporal resolution $\Delta$ by dividing the whole data temporal window of length $T$ into $T/\Delta$ time intervals, and we build on each interval a binary adjacency matrix $A_t$ with elements $A_{ij,t}$ equal to 1 if there is at least one interaction between $i$ and $j$ during $(t - \Delta, t]$ and zero otherwise.

In the temporal fitness model, each node $i$ is assigned an intrinsic variable that we call "activity" level, $a_i \in (0, 1]$, and the probability $u$ that nodes $i$ and $j$ interact (e.g., through a face-to-face contact, a bilateral financial transaction, etc.) during any given time interval is simply given by the product of their activity levels[34]:

$$u(a_i, a_j) = a_i a_j, \tag{1}$$

In a static network context, this class of network model is called the fitness model and has been used to model network generative processes[35–37]. The null model we obtain is thus a sequence of successive independent realizations of the (static) fitness model. It settles a baseline of how much two nodes are expected to interact, given their activities, if interaction partners are selected at random at each time step. In other words, we do not assume any underlying pre-existing network structure, and we consider a hypothetical situation in which there is always a positive chance of interaction between any two nodes. Note also that this temporal null model does not contain any a priori knowledge of the group structure of the nodes. An interesting modification could be to superimpose group labels or node properties (e.g., gender or age for nodes representing individuals) and interaction probabilities depending on the nodes' properties.

In the simplest version of this null model, we consider constant activity values for each node. In addition, we present in Supplementary Note 1 a refined version that takes into account temporal variations of the overall interaction activity in the system. This is achieved through the introduction of a time-varying parameter $\xi(t)$. In that case, the null model is defined by the fact that the probability of nodes $i$ and $j$ establishing a connection at time $t$ is $u(a_i, a_j, t) = a_i a_j \xi(t)$. We present here the case of constant $\xi(t) = 1$, as we can then obtain an analytical form for the probability distribution of the number of interactions of a pair of nodes, while only an approximate formula is available in the more general case. Note that this number is at most $\tau = T/\Delta$, i.e., the number of time intervals given the resolution $\Delta$. We show in Supplementary Note 1 that both methods yield quite similar results in the cases studied here (see Supplementary Figure 4).

**Significant ties.** To uncover significant ties with respect to the null model described above, we proceed in two steps (Fig. 1). First, given a data set, we estimate the node activity levels $\mathbf{a} \equiv (a_1,...,a_N)$, within the temporal fitness model. Note that the activities $\{a_i\}$ are latent variables that rule the probabilities of interactions between nodes in the model, but are neither directly observable nor inferred from the local information about the nodes in a data set. They can, however, be estimated using a maximum likelihood estimation, as described in Methods, yielding the values $\mathbf{a}^* \equiv (a_1^*, \dots, a_N^*)$. From the model's definition, the $a_i^*$ are expected to be correlated with the total number of interactions with other nodes, and we show indeed in the Supplementary Note 1 that these estimated activity parameters are proportional to the strengths of the nodes and correlated with their degree.

We then compute for each interacting pair of nodes $i$ and $j$ the probability distribution of their total number of interactions $m_{ij}$ in the null model, which is given by the following binomial distribution:

$$g(m_{ij}|a_i^*, a_j^*) = \binom{\tau}{m_{ij}} u(a_i^*, a_j^*)^{m_{ij}} (1 - u(a_i^*, a_j^*))^{\tau - m_{ij}}. \tag{2}$$

Let $m_{ij}^c$ denote the $c$-th percentile ($0 \le c \le 100$) of $g(m_{ij}|a_i^*, a_j^*)$, i.e., $c/100 = G(m_{ij}^c|a_i^*, a_j^*)$, where $G$ is the cumulative distribution function (CDF) of $g(m_{ij}|a_i^*, a_j^*)$, namely $G(m_{ij}^c|a_i^*, a_j^*) = \sum_{m_{ij}=0}^{m_{ij}^c} g(m_{ij}|a_i^*, a_j^*)$. If the actual empirical number of interactions $m_{ij}^o$ between $i$ and $j$ is larger than $m_{ij}^c$, it means that this empirical number cannot be explained by the null model at significance level $\alpha \equiv 1 - c/100$: this indicates that $i$ and $j$ are connected by a significant tie. The $P$-value of the test is given by $1 - \sum_{m_{ij}=0}^{m_{ij}^o} g(m_{ij}|a_i^*, a_j^*)$. For a given significance level $\alpha$, we can test the significance of the set of interactions composing each tie independently from the other ties[17]. Note that, even if the significance of a tie is determined from an aggregated number of interactions, a significant tie does not correspond here to a static

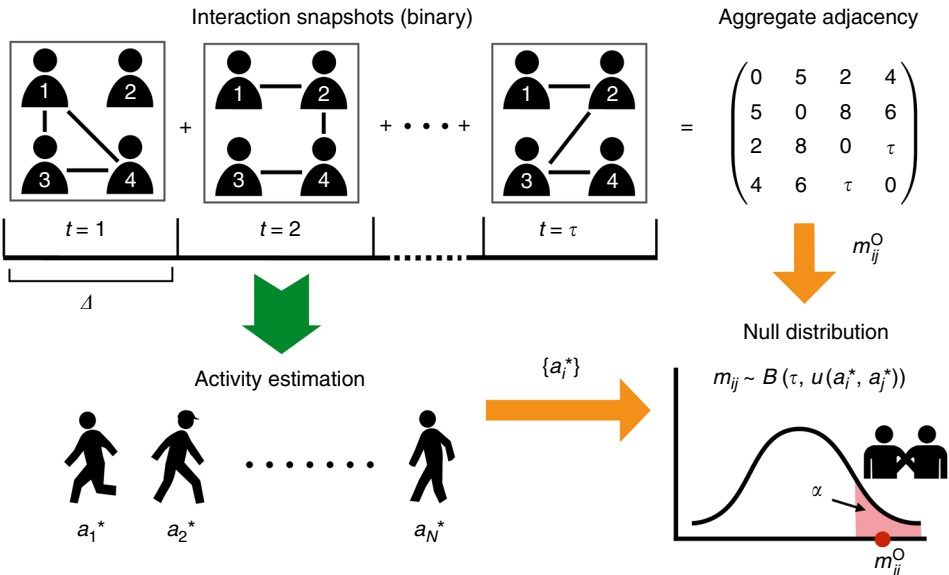

**Fig. 1** Sketch of the filtering method. From the temporal network at resolution $\Delta$, described by $\tau = T/\Delta$ adjacency matrices, we estimate the set of node activities $(a_1^*, \dots, a_N^*)$, and thus the probability distribution of the number of interactions between any pair of nodes $(i, j)$ under the null model. We compare the empirical value $m_{ij}^o$ with the percentiles of this distribution to determine the significance of the pair $(i, j)$'s interactions

edge but to an interacting pair of nodes with their set of temporally resolved interactions. In particular, the backbone given by all interactions in the significant ties remains a temporal network. Tuning $\alpha$ allows us to probe more and more significant pairs by decreasing $\alpha$, and/or to tune the number of ties retained in the backbone, providing a systematic filtering method that we call Significant Tie (ST) filter.

Thanks to the use of the null model, a pair of interacting nodes can be significant even if their number of interactions is small, as long as their individual activity levels are sufficiently low. Reciprocally, ties with a large number of interactions might not be significant if the two involved nodes are very active. The ST filter controls indeed for the difference between nodes in terms of intrinsic activity levels. As a consequence, the significant ties identified by our method are "irreducible" in the sense that their significance cannot be attributed to local node-specific properties[17], such as the node degree and strength in the aggregated network. The probability of interaction between two nodes under the null hypothesis is indeed determined by an interplay of global and local information through the maximum likelihood estimation (see Methods). The resulting network of interactions between the significant pairs of nodes may thus be regarded as an irreducible backbone of the temporal network under study[17]. The MATLAB code for the ST filter is available from the Zenodo website https://doi.org/10.5281/zenodo.1243994

**Going beyond significant ties with significant temporal structures.** Using a temporal fitness model as a null model allows us to go beyond the usual tests concerning the significance of ties, and to assign a significance to higher-order structures, such as temporal motifs. To illustrate this point, let us consider the simple case of a triadic interaction between three nodes $i$, $j$, and $k$; the empirical number of time intervals in which the three pairs $(i, j)$, $(j, k)$, and $(i, k)$ are simultaneously interacting, denoted by $r_{ijk}^o$, can be compared with the probability distribution of the number $r_{ijk}$ of occurrences of such triangles in the null model. For each time interval, the probability that $i$, $j$, and $k$ are forming a triangle of interactions in the temporal fitness model is

$$v(i,j,k) = u(a_i^*, a_j^*) \cdot u(a_j^*, a_k^*) \cdot u(a_k^*, a_i^*), \quad (3)$$

so that $r_{ijk}$ obeys the following probability distribution in the null model:

$$h(r_{ijk}|a_i^*, a_j^*, a_k^*) = \binom{\tau}{r_{ijk}} v(i,j,k)^{r_{ijk}} (1 - v(i,j,k))^{\tau - r_{ijk}}. \quad (4)$$

Similarly to the case of dyads, we define for each significance level $\alpha = 1 - c/100$ the significant triads as those such that $r_{ijk}^o$ is larger than the $c$-th percentile of $h(r_{ijk}|a_i^*, a_j^*, a_k^*)$.

Note that this method can easily be generalized to any set of temporally constrained interactions (e.g., occurring in a sequence of successive snapshots) or motifs. On the contrary, any filtering method based directly on the aggregated network, and not taking into account the temporality of the data, is by construction unable to define a null hypothesis for simultaneous interactions (or interactions with a given temporal sequence) and thus to assign a significance to such patterns.

**Validation and case studies.** We now apply the ST filter to both synthetic and real data sets, and compare its outcome with other filtering methods. On the one hand, we consider two methods that use directly the static, temporally aggregated network, namely the disparity filter (DP filter)[13] and the enhanced configuration model (ECM filter)[17], whose computations are recalled in Supplementary Note 2. In addition, we also examine two methods that partially take into account the temporal nature of interactions by implementing a static filter on each temporal snapshot. Specifically, we apply the ECM filter (respectively, the DP filter) on each snapshot, and a pair of nodes is regarded as significant if an edge between the two nodes is identified as significant in at least one snapshot: this defines two baseline temporal filters that we call, respectively, ECM-R (ECM-repeated) and DP-R (DP-repeated).

Let us first consider as a validation exercise a synthetic temporal network with known properties, composed of a superposition of random and strong edges. We consider $N = 300$ nodes, each node $i$ endowed with an internal variable $a_i' \in [0, 1]$ drawn from a Beta distribution, and nodes $i$ and $j$ are connected at each time step with probability $a_i' a_j'$. We then

superimpose to this random temporal structure additional temporal edges during a time window of length $T$: we first select at random 20% of the interacting pairs, and we add interactions for these pairs, so that they become a known set of "strong" ties. We then treat these synthetic data as the other data sets: we create a sequence of $\tau = T/\Delta$ network snapshots by aggregating all the interactions made over time windows of length $\Delta$. More details about the generation of synthetic networks are provided in Supplementary Note 3 (see also Supplementary Figure 7).

Figure 2a shows that the fraction of significant edges detected by all the filters we consider decreases as $\alpha$ decreases and lies below the fraction of known strong pairs (namely 0.2) as soon as $\alpha$ takes reasonable values such as $\alpha < 0.01$. The fraction of false positives is then negligible in all cases (see Supplementary Figure 5): almost all ties detected as significant correspond to strong edges in the synthetic data (for $\alpha < 0.01$). Figure 2 highlights two interesting features. First, the ST filter is much more successful in detecting strong ties compared with the other filtering methods examined as soon as $\alpha$ decreases below 0.01. Second, both the average and the distribution of the fraction of strong edges detected by the ST filter are stable for a broad range of $\alpha$. This stability shows that the significant ties can be retrieved without the need for fine-tuning the significance level, and that

the ST filter retrieves the significant ties in a robust fashion. On the contrary, the fraction of detected strong ties decreases very fast for the other filters when the significance level increases. We show in Supplementary Figure 6 that ST in particular identifies much better than the other filters the significant ties with small weights. Note that no filter detects all strong ties. This is due to the fact that some of the node pairs selected to be "strong" ties connect nodes with large activity values: the additional interactions might then yield an overall temporal sequence that is still compatible with the null model, i.e., their number of interactions could still be explained by chance, given their activity levels. This means overall that it is reasonable to regard the fraction of ST edges as a conservative measurement for the fraction of significant ties in a data set.

Note that we also consider in Supplementary Note 3 and Supplementary Figure 8 a different synthetic network model, which also highlights how ST and the other filters differ in the type of ties they detect as significant. Let us now turn, however, to the empirical data case studies. We present in the main text the main results for a subset of the data sets considered and refer to the Supplementary Figures 10–20 for the other data sets.

Figure 3 first displays the number of significant ties as a function of the significance level $\alpha$, for the five methods. As $\alpha$

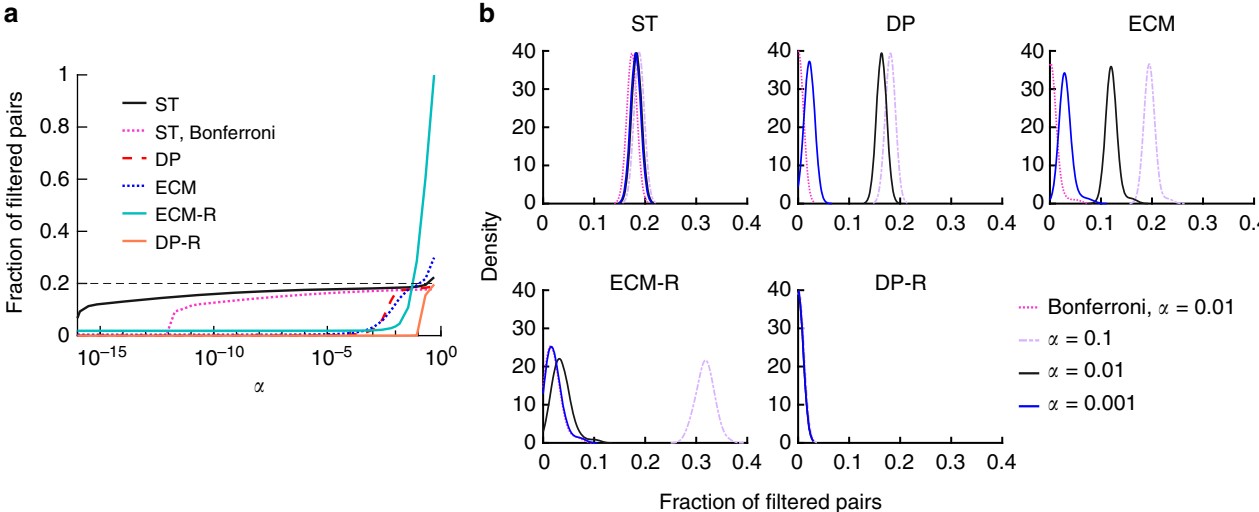

**Fig. 2** Fraction of significant pairs detected in synthetic temporal networks. Synthetic networks are generated such that exactly 20% of edges have strong ties. We generate 100 networks with $N = 300$, $T = 300$, and we use $\Delta = 10$ (see Supplementary Note 3 for a detailed description of synthetic networks). **a** Average fraction of significant ties as a function of the significance level. The dotted pink line denotes ST edges with Bonferroni correction for multiple tests. **b** Distribution of the fraction of significant pairs for different significance levels

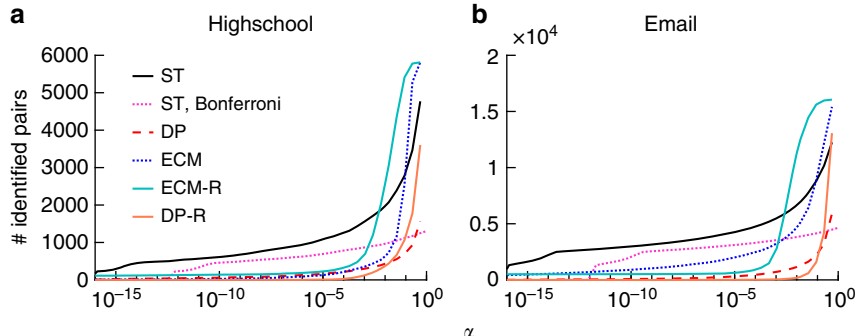

**Fig. 3** Number of significant ties as a function of the significance level for the **a** Highschool and **b** Email data sets. The dotted pink line denotes the ST filter with Bonferroni correction, while black solid, red dashed, blue dotted, light-blue solid, and orange solid lines represent the ST, DP, ECM, ECM-R, and DP-R filters without Bonferroni correction, respectively. Temporal resolution is $\Delta = 15$ min for Highschool and 1 day for Email

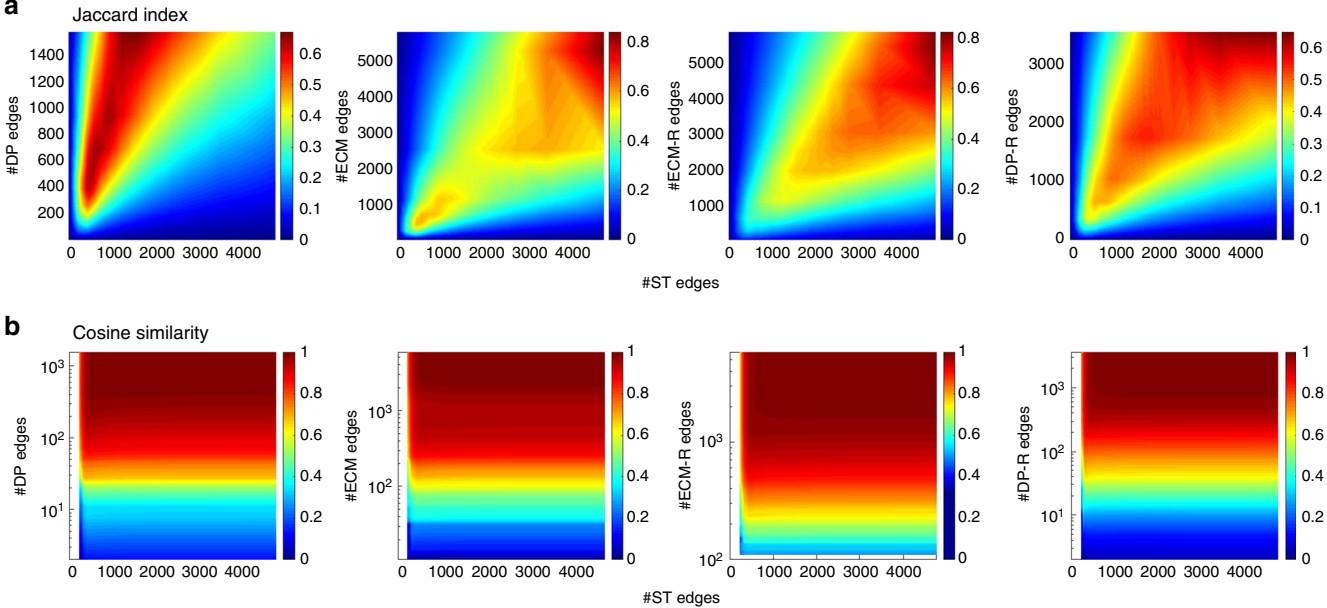

**Fig. 4** Comparison between the sets of edges detected as significant by different filters, for the Highschool data. **a** The Jaccard index is defined by $J(I_{ST}^{\alpha}, I_x^{\alpha'}) = |I_{ST}^{\alpha} \cap I_x^{\alpha'}| / |I_{ST}^{\alpha} \cup I_x^{\alpha'}|$, where $x \in \{DP, ECM, ECM-R, DP-R\}$. We plot the Jaccard index versus the numbers of edges detected at different significance levels, ranging from $\alpha = 10^{-17}$ to 0.5 (obviously, the higher the value of $\alpha$, the larger the number of significant edges). **b** Same for the cosine similarity defined in Eq. (5)

decreases, this number decreases for all methods. Interestingly, however, the number of significant ties remains much larger for our method as soon as $\alpha$ enters a regime of high statistical significance (e.g., $\alpha < 10^{-2}$), and decreases only slowly with $\alpha$. As $\alpha$ becomes very small, i.e., at very high statistical significance, DP and ECM (as well as the DP-R and ECM-R filters) retain only a very small number of edges, while the ST filter still uncovers a relatively large number of significant node pairs. We also present the results for the ST filter with Bonferroni correction, in which the significance level is adjusted by dividing by the number of edges to control for false positives. The resulting backbone of interactions between significant pairs at very low $\alpha$ (e.g., $\alpha$ between $10^{-10}$ and $10^{-5}$) is very stable for the ST filter, with a size decreasing only slowly with $\alpha$, and might thus be regarded as a fundamental backbone of the data set, a feature not obtained with the other filters. See Supplementary Figure 10 for the similar results obtained with other data sets and different parameters.

Given the differences in the definition of the various filters, it is important to understand to what extent these filters select distinct or similar sets of ties. We first quantify the similarity between backbones obtained by different filters through the Jaccard index $J(I_{ST}^{\alpha}, I_x^{\alpha'}) = \frac{|I_{ST}^{\alpha} \cap I_x^{\alpha'}|}{|I_{ST}^{\alpha} \cup I_x^{\alpha'}|}$. It gives the fraction of common edges between the backbone obtained by the ST filter at significance level $\alpha$ and another backbone $x \in \{DP, ECM, DP-R, ECM-R\}$ at significance level $\alpha'$. A Jaccard equal to 1 means that both methods yield the same exact set of edges, while $J = 0$ means that the backbones are disjoints. As the different methods yield very different backbone sizes for a fixed significance level, we show in Fig. 4a a color plot of the Jaccard index as a function of the number of node pairs retained by each filtering method. In all cases, the largest Jaccard indices are obtained when the number of edges are similar: they reach at most ~80% when $\alpha$ takes a meaningful value (e.g., $\alpha < 10^{-2}$) and decrease as the backbone size decreases (Supplementary Figure 11). This shows that the backbones obtained by different methods show some similarity but are not equivalent.

To investigate this in more detail, we also consider a weighted measure of the similarity, as the Jaccard index does not take into

account that different ties can correspond to very different number of interactions (i.e., weights). We thus compute the cosine similarity between the backbones obtained by different filters:

$$\sigma(x, x') = \frac{\sum_{i<j} w_{ij}^x w_{ij}^{x'}}{\sqrt{\sum_{i<j} \left(w_{ij}^x\right)^2} \sqrt{\sum_{i<j} \left(w_{ij}^{x'}\right)^2}}, \quad (5)$$

where x encodes both the filtering method (ST, DP, ECM, DP-R, and ECM-R), and the significance level $\alpha$ and the sums run only on the pairs of nodes present in the backbones. Figure 4b indicates that values larger than the Jaccard index are obtained, with similarities of 0.9–1, decreasing below 0.8 only when the backbone sizes becomes small. The various backbones seem thus to all retain similar sets of ties with large weights, while differing when assessing the significance of pairs of nodes with smaller numbers of interactions. Similar results are obtained with all data sets (Supplementary Figure 12).

We explore further in Supplementary Figure 13 the distributions of weights (i.e., of the number of interactions) of the ties considered either as significant or not by different filters. Significant ties display on average larger weights than the nonsignificant ones. The DP filter in particular is closer to a thresholding procedure (i.e., selecting edges with high weights) than the other filters, with only a narrow range of weight values for which both significant and nonsignificant edges can be found (we have checked that the Kullback–Leibler divergence between the weight distribution obtained with a given filter and the one of a global thresholding is the smallest in the case of the DP filter). This is in agreement with the result noted in ref. [17] that this filter tends to retain larger weights. For all the other filters, the weight distributions of the significant edges are quite similar and, most importantly, are as broad as the original weight distribution of the whole network. Thanks to the use of null models, these filters manage indeed to find significant ties at all intensity scales,

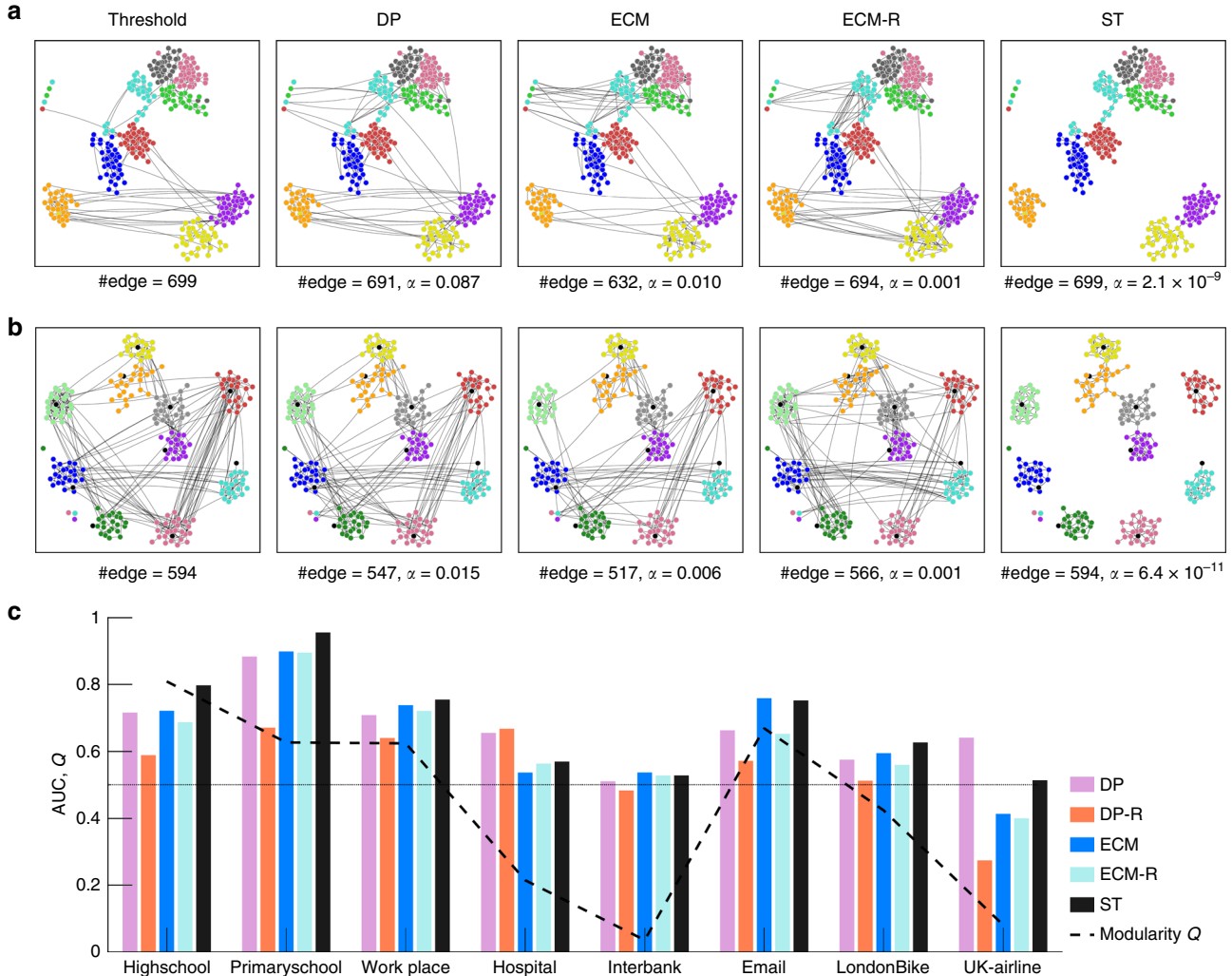

**Fig. 5** Backbones and community structure. Visualization of the backbones obtained by different filtering methods, at similar numbers of edges, for the **a** Highschool and **b** Primaryschool data sets. The nodes are shown in the same position for all the backbones of a given data set. "Threshold" corresponds to a simple thresholding procedure on the aggregated network. Different colors denote different classes. For the Primaryschool data, black circles represent the teachers. The ST filter detects a larger fraction of intra-class edges than the other filtering methods. **c** AUC of the ROC curve (see Supplementary Figure 9 for the ROC curves) for the identification of intra-community edges. The black dashed curve gives the weighted modularity $Q$[42,43] calculated by regarding the actual groups (classes for Primaryschool and Highschool, departments for Workplace, and roles for Hospital) as communities in the original aggregate networks. For the Interbank data set, banks are classified into Italian banks and foreign banks. For Email, LondonBike, and UK-airline, the communities are detected by Infomap[44]

i.e., with a wide range of numbers of interactions. For instance, for the ST filter, there exists a significant tie at $\alpha = 0.01$ for the Workplace data set with only a single interaction $\left( m_{ij}^{o} = 1 \right)$, and in other data sets, some significant ties have $m_{ij}^{o} = 2$ or 3 only.

We finally investigate the effect of varying the temporal resolution $\Delta$ in Supplementary Figure 15. An increase in $\Delta$ (i.e., lower resolution) generally lowers the number of significant ties (Supplementary Figure 15a). This results in a decrease of the Jaccard index between the sets of significant ties as the difference in temporal resolution increases (Supplementary Figure 15b). However, Supplementary Figure 15c shows that this decrease is mainly caused by the fact that some significant pairs are no longer detected as $\Delta$ increases, while ties detected as significant at a lower resolution $\Delta'$ are also detected as significant at a higher resolution $\Delta$ ($\Delta < \Delta'$).

In several of the data sets we consider, nodes can be classified into different groups, corresponding, e.g., to classes in schools, departments in the workplace, and roles in the hospital. In the Highschool, Primaryschool, and Workplace cases, these groups

define a clear-cut community structure[5,26,27], while nodes from different groups are more mixed in the Hospital data set[28]. This is confirmed by the values of the weighted modularity of the partition corresponding to these groups shown in Fig. 5c and Table 1. We also obtain a clear community structure for the Email and Londonbike cases.

Figure 5a, b shows a visualization of the backbones obtained by the different filtering methods for the Highschool and Primaryschool data sets. These visualizations indicate that the backbone obtained by the ST filter seems to separate the network into connected components corresponding to these communities more efficiently than the other filters, at fixed number of edges. We show that this is indeed the case through two quantitative indicators. First, we measure, as a function of the backbone size, the fraction of intra-group edges (Supplementary Figure 14). It is larger than the random baseline (in which edges are kept completely at random) for all filters, approaching one as the number of edges decreases, and maximal for the ST filter. Second, we consider each filtering method as a prediction task for finding

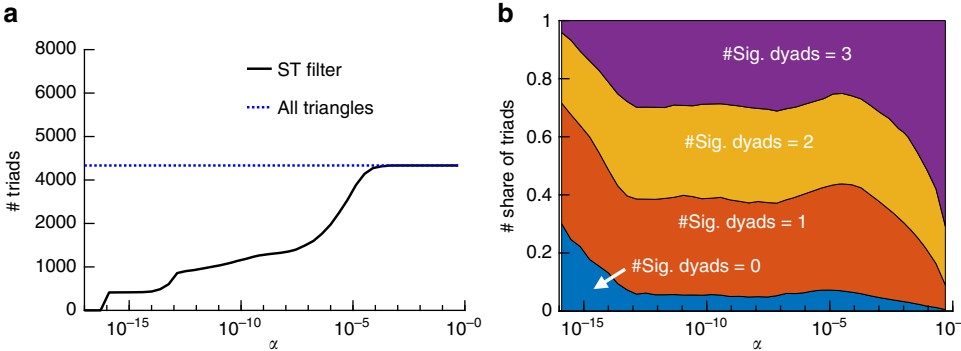

**Fig. 6** Significant triadic relationships in the Highschool data set, for a temporal resolution $\Delta = 1$ min. **a** Number of significant triads as a function of $\alpha$. **b** Fraction of significant triadic relationships with a given number of significant dyads, as a function of $\alpha$

intra-group edges. We use $\alpha$ as a parameter and measure, for each $\alpha$, the true and false positives (edges in the backbone that are/are not intra-group edges) and true and false negatives (edges not in the backbone that are not/are intra-group edges). Varying $\alpha$ allows us to build a ROC curve for each filter (see Supplementary Note 4 for details), and we find that the area under the curve (AUC) of the ROC curve obtained for the ST filter is higher than for the other filters for the five data sets for which the modularity is high.

The fact that the ST backbone selects mostly intra-community node pairs suggests that inter-community interactions may be explained by the null model of random interactions ruled by the node activity levels. Inter-community edges, which act as bridges, play an important role in propagating information, spreading ideas, and diffusion of influence[24,38]. Nevertheless, our analysis shows that the actual weights of these edges are statistically indistinguishable from the ones resulting from interactions at random times, regardless of the history of interactions between the two nodes, once the node activities are given. This hints at a way to represent the original temporal network as a superposition of (i) a backbone of significant ties and (ii) connections extracted at random according to the null model between nodes of different groups. Such a representation would yield a refined version of the contact matrix of distributions put forward in refs. [39,40].

While the ST filter detects intra-community edges more efficiently than the other filters do, our results show that the other filters also tend to detect intra-community edges when there is an explicit community structure. We investigate this further in Supplementary Figure 16 in which we show that the intra-community edges of a node are more likely to be significant compared with its inter-community edges, and this for any node degree. This property is common to all filters but most evident in the ST case. This suggests that the non-significance of inter-community edges is not explained by differences in aggregate degree of their end nodes, but rather reflects an intrinsic difference between intra- and inter-community edges. The ST filter is able to exploit such an intrinsic difference most efficiently.

As described above, we can use the temporal null model defined above to extract significant temporal structures. We illustrate this in the case of triads with a significant number of simultaneous interactions. Note that such a task is by construction impossible in the filtering methods defined on static aggregated networks, as no temporal constraint can be detected: for instance, a triangle $(i, j, k)$ in an aggregated network can in fact result from interactions between the pairs of nodes $(i, j)$, $(j, k)$, and $(k, i)$ occurring at different times. Moreover, the null models for the DP and the ECM (and thus the DP-R and ECM-R)

filters are designed to detect significant dyads, and we cannot directly exploit their null distributions to test the significance of triads or other structures. Therefore, the only way to define a structure as significant for these filters is to impose that all the ties of the structure are significant. Note that the case of the DP-R and ECM-R filters is particular: these filters correspond to applying the DP (resp. the ECM) to each snapshot, so that we can in this case also define the significance of temporal structures. However, this is done in a somehow trivial way that is not grounded in a temporal null model. For instance, we define a simultaneous triad as significant for the ECM-R or DP-R if there is at least one snapshot in which its three edges coexist and are all three significant.

We show in Fig. 6a the number of significant triadic relationships as a function of the significance level $\alpha$, for the Highschool data set and temporal resolution $\Delta = 1$ min. Similar results are shown in Supplementary Figure 17 for the other data sets and different values of $\Delta$. For filtering levels $\alpha > 10^{-4}$, almost all the triangles present in the aggregated network are considered as significant, except for the Interbank and UK-airline data. However, the number of significant triadic relationships decreases as $\alpha$ becomes lower, determining a set of triads such that their number of simultaneous interactions cannot be explained by the temporal null model and the individual nodes' activity levels. We show in Supplementary Figure 18 the number of significant simultaneous triads in the ECM-R filter: it decreases very fast as the significance level increases, showing that the ST filter is more able to detect such structures.

Figure 6b highlights moreover a striking feature of the significant triads detected by our temporal null model, and absent by definition in the DP-R and ECM-R cases, namely, that they do not necessarily correspond to three significant ties. In fact, the number of significant ties in a significant triad can take any value between 0 and 3 (see also Supplementary Figure 19). Reciprocally, not all triangles made by three significant ties turn out to be significant triads (Supplementary Figure 20). This clearly shows how the temporal null model allows us to go beyond the definition of significant ties and find significant higher-order structures that could not be unveiled by a static approach nor by their too simple generalizations. Indeed, considering triangles made by significant ties does not guarantee that the corresponding triads have (a significant number of) simultaneous interactions, while on the other hand, ties $(i, j)$ with a nonsignificant number of interactions when considered as dyads can turn out to interact a significant fraction of times simultaneously with two other dyads $(j, k)$ and $(i, k)$ for a certain $k$, forming thus a significant triad.

## Discussion

In this paper, we have presented a method to find significant ties and structures in temporal network data sets, taking into account for the first time their temporal nature. To this aim, we have defined a null model of interactions that takes into account the heterogeneity in the activity of individual nodes and the temporal dimension of the system, and can potentially be extended to include temporal variations in the overall network activity, due, for instance, to circadian or weekly rhythms, imposed schedule constraints, etc. We compute for each pair of nodes the distribution of their number of interactions in the null model, and compare the empirical value to this distribution. For any chosen significance level, we thus define as significant pairs of nodes those with a number of interactions that cannot be explained by the null model. As the null model includes the heterogeneous activity of nodes, the temporal network backbone composed by the ties with significant numbers of interactions is not reducible to the nodes local properties and contains ties with a broad distribution of numbers of interactions. Varying the significance level allows us to tune the number of node pairs in this backbone.

We have compared the results obtained with our method with other backboning methods built for a weighted static network, hence applied here on the temporally aggregated network and with two baseline temporal extensions of static filters. This reveals interesting similarities and differences and highlights the interest and impact of our filtering method. Our method yields, at a given significance level, more pairs of nodes than the other filters. In particular, we have shown in a synthetic benchmark case that our filter identifies correctly more ties known a priori to be significant. A more detailed comparison shows that the difference does not come from ties with large numbers of interactions, which are similarly selected by all methods (and easily found even by a trivial global thresholding), but rather by the fact that our filter uncovers more significant pairs of nodes with a small number of interactions, more difficult to identify. The resulting distribution of weights of the significant ties is broad, showing the ability of the ST filter to detect significant ties at all scales. Moreover, an interesting feature of the ST filter resides in its relative stability for broad ranges of the significance level, while the number of significant ties is more sensitive to the significance level for other filters. This makes it possible to define a backbone in a more robust way, without fine-tuning the significance level, and, in the synthetic case, to retrieve also in a more robust way the set of significant ties. We have also observed that, for networks with a strong community structure, our method tends to uncover mostly intra-community ties, showing that the weights of the inter-community ones, once the node activities are given, can be explained by random interactions ruled by these activities.

The temporal nature of the null model used to define the ST filter has also another crucial consequence: it allows to go beyond the concept of significant ties and to identify significant temporally constrained structures, not necessarily composed only of significant ties. Such endeavor is by definition impossible with static filters, and even the ECM-R and DP-R filters, which take into account some temporality, can identify temporal structures only in a trivial way, by defining them as sets of significant ties. We have illustrated this interesting advantage of our method in the case of sets of simultaneous interactions, which have a clear importance in social terms (it is clearly not the same to have three interactions $(i, j)$, $(j, k)$, and $(i, k)$ between three individuals at the same moment or at different times), but also for processes such as epidemic spread occurring on top of a temporal network[41]. We have in particular shown that significant triads of simultaneous interactions are not equivalent to triangles of three significant dyads, illustrating the need to take into account the temporality of

these structures, which could not be uncovered by static filtering methods.

Our work has some limitations worth mentioning. In particular, it does not consider individual temporal variations in node activity, nor possible insertion or deletion of nodes into/from the system. Moreover, even if the null model is temporal, and the resulting backbone is as well a temporal network, the significant ties are defined as whole sets of interactions between nodes, while a finer resolution might be desirable to distinguish significant events within each tie.

Finally, our work hints at several perspectives and future research directions. First, it would be interesting to refine data representations such as the ones put forward in refs. [39,40], by combining a backbone at a certain significance level and a contact matrix representing in a summarized fashion the nonsignificant ties. Another interesting avenue would be to represent the data as a backbone plus the set of activities $\{\{a_i^*\}, \xi(t)\}$. Both possibilities are made more achievable by the stability of the set of significance ties over a broad range of significance values, as obtained in the ST filter. In both cases, these representations should be validated by numerical simulations of various types of processes on top of the data. The relevance of such representations is twofold: on the one hand, they allow to summarize and generalize complex data sets in a way that can be fed into data-driven models of dynamical processes such as epidemic or information spreading; on the other hand, they keep the minimum amount of detailed information on the precise interactions, summarizing less relevant details as distributions or averages, and thus in a way that might be easier to render data anonymous. Finally, another direction of research would be to define a backbone at a finer resolution, namely that would be composed of (sets of) significant interactions instead of ties or set of ties.

## Methods

**Data processing**. Different filtering methods use different network formats. Here, we summarize the data-processing procedure.

- Highschool, Primaryschool, Workplace, Hospital: For the ST filter, the network snapshots (i.e., unweighted adjacency matrices) are created so that each represents interactions between people observed over a $\Delta$-minute interval ($\Delta = 15$ unless otherwise noted). We exclude the time interval between the last contact of a day and the first contact of the following day. The aggregate network for the DP and ECM filters represents all the interactions recorded over the whole data period, in which edge weights are given by the total numbers of interactions. For the DP-R and ECM-R filters, we use a sequence of weighted networks aggregated over the time windows of duration $\Delta$, in which edge weights represent the numbers of interactions observed over a $\Delta$-minute interval.

- Interbank: Snapshots for the ST filter are given by daily networks formed by overnight bilateral transactions between banks between June 12, 2007 and July 9, 2007 (20 business days). The daily networks are unweighted and undirected. The edges of the aggregate network for the DP and ECM filters are weighted by the number of transactions observed over the data period. For the DP-R and ECM-R filters, a weighted network is created for each day so that the weight of each edge represents the number of intraday transactions on the corresponding day.

- Email: The edges of the network snapshots for the ST filter represent the presence of emails between two members of a research institution in the EU on a given day. The edges of the aggregate network for the DP and ECM filters are weighted by the total numbers of emails over the data period. For the DP-R and ECM-R filters, we use a sequence of daily weighted networks, in which edge weights represent the numbers of emails exchanged on a given day.

- LondonBike[31]: The data contain all the trips taken between 00:00 and 23:59 on June 22, 2014 (with a time resolution of 1 minute). For the ST filter, each network snapshot represents the trips between stations started within a 15-minute interval. The aggregate network for the DP and ECM filters represents all the trips recorded over the whole day, in which an edge weight is given by the total number of trips between two stations. For the DP-R and ECM-R filters, we use a sequence of weighted networks, in which edge weights represent the numbers of trips observed in a given 15-minute interval.

- UK-airline[32]: Each of the snapshots for the ST filter is an unweighted and undirected adjacency matrix of domestic airlines in the United Kingdom in a given year. Edges of the aggregate network for the DP and ECM filters are

weighted by the number of snapshots in which the edge is present, over the whole data period (1990–2003). For the DP-R and ECM-R filters, we use as snapshots the yearly weighted networks whose edge weights are given by the number of passengers recorded over the corresponding year, because the number of flights in a given year is not available from the data.

**Estimation of nodal activity**. We perform a maximum likelihood (ML) estimation of $\mathbf{a} \equiv (a_1, \ldots, a_N)$, taking the $\tau$ temporal snapshots as input, where $\tau = \lfloor T/\Delta \rfloor$. If two individuals are independently matched in each time interval according to probability $u(a, a')$, then the number of times temporal edges are formed between nodes $i$ and $j$ over $\tau$ time intervals is a random variable $m_{ij}$ that follows a binomial distribution with parameters $\tau$ and $u(a_i, a_j)$. Therefore, the joint probability function leads to

$$p(\{m_{ij}\}|\mathbf{a}) = \prod_{i,j:i\neq j} \binom{\tau}{m_{ij}} u(a_i, a_j)^{m_{ij}} (1 - u(a_i, a_j))^{\tau - m_{ij}}, \quad (6)$$

where $m_{ij} \leq \tau$ denotes the count of temporal edges between $i$ and $j$ observed over $\tau$ periods in the null model. The log-likelihood function for the empirical data $\{m_{ij}^o\}$ is thus given by

$$\mathcal{L}(\mathbf{a}) = \log p(\{m_{ij}^o\}|\mathbf{a}) = \sum_{i,j:i\neq j} \Big[ m_{ij}^o \log (a_i a_j) + (\tau - m_{ij}^o) \log (1 - a_i a_j) \Big] + \text{const.}, \quad (7)$$

where "const." denotes the terms that are independent of $\mathbf{a}$. Note that the sum runs over all pairs of nodes $(i, j)$, including those with $\{m_{ij}^o\} = 0$. The ML estimate of $\mathbf{a}$ is the solution for the following $N$ equations:

$$H_i(\mathbf{a}^*) \equiv \sum_{j:j\neq i} \frac{m_{ij}^o - \tau a_i^* a_j^*}{1 - a_i^* a_j^*} = 0, \forall i = 1, \ldots, N, \quad (8)$$

The first-order condition (8) is obtained by differentiating the log-likelihood function (7) with respect to $a_i$. The system of $N$ nonlinear equations, $H(\mathbf{a}) = \mathbf{0}$, can be solved by using a standard numerical algorithm. (We solve the equation by using MATLAB function fsolve, which is based on a modified Newton method, called the trust-region-dogleg method. The initial values of $a_i$ are given by the configuration model: $a_i = \sum_{j:j\neq i} (m_{ij}^o/\tau) / \sqrt{2 \sum_{i<j} m_{ij}^o/\tau}$, where the numerator and the denominator represent the means of $i$'s temporal degree and the doubled number of total temporal edges, respectively.). The obtained ML estimates of $\mathbf{a}$ are denoted by $\mathbf{a}^* \equiv (a_1^*, \ldots, a_N^*)$. The numbers of contacts obtained from the model and the empirical data are compared in Supplementary Figure 1 in Supplementary Note 1. See also Supplementary Figures 2 and 3 for additional investigations into the correlation between activity and aggregated node properties. The extension of the method to include time-varying probabilities of creating interactions is shown in Supplementary Note 1.

**Code availability**. The code is available from the Zenodo website https://doi.org/10.5281/zenodo.1243994.

## Data availability

Seven of the eight data sets we use are publicly available. One is commercially available form a third party. The four SocioPatterns data sets were downloaded from http://www.sociopatterns.org/datasets. The Interbank data set is commercially available from http://www.e-mid.it. The Email data set was downloaded from the SNAP repository http://snap.stanford.edu/data/index.html. The LondonBike data were downloaded from https://github.com/konstantinklemmer/bikecommclust. The UK-airline data set was downloaded from http://www.bifi.es/~cardillo/data.html.

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

## Acknowledgements

TK acknowledges financial support from the Japan Society for the Promotion of Science Grants no. 15H05729 and 16K03551.

## Author contributions

A.B., T.K. and T.T. designed the study. T.K. performed the numerical analysis. A.B., T.K. and T.T. wrote the paper.

## Additional information

**Competing interests:** The authors declare no competing interests.

