## [Peer Review File · Nature Communications]

Reviewers' comments:

Reviewer #1 (Remarks to the Author):

Authors of this manuscript propose a method to identify significant links in temporal networks, this way constructing the “back bone” of the underlying structure. As the authors underlines, there is a need for the development of such methods for the better analysis and reduction of recently available large temporal interaction datasets. In this sense the paper is timely and important. The manuscript is well written and the proposed methodology, as introduced, is scientifically sound. However, I would like to ask the authors to address my various comments listed below. Some of the comments may be due to my misunderstanding, thus their clarification should be easy, while some other comments may highlight important conceptual points which needs to be addressed.

Major comments:

- One of the cornerstone quantity of the proposed method is node activity. However, the authors miss to define what node activity means in case of real temporal networks. Is it the total number of observed interactions? Is it the number of interactions per time window? Activity appears as a parameter in Eq.1 and in the definition of the maximum likelihood estimator but it should be introduced earlier for better understanding.

- My main concern about the paper might be a confusion. The aim of the paper is to filter links from an already existing structure by deciding whether interactions on a link appear more frequently than one would assume solely from the activity levels of the two ending nodes. My first question is why the authors compute the null model between every pairs of nodes in the network? One does not need to filter non-existing links, on the other hand defining the likelihood this way may give the impression that a fully connected structure is considered in the definition.

- I understand that the method focuses on dyadic interactions in the computation of the backbone, while neglecting any aggregated network information, e.g., node degree. However, degree is a very important property in this setting as it determines the number of neighbors with whom a node needs to “share” its own activity. For example, significance of a link may be very different for an less active node with small degree than for an less active node with high degree. Since the aim here is not to filter temporal networks “on the run”, but the dynamics of an already observed network, the aggregated degrees are already known in advance. This way considering degrees in the null model definition might be possible.

- Related to my previous comment, considering no degree information, inter-community links naturally appear mostly insignificant. Inter-community links are typically maintained by small number of interactions, however, they may be significant for a node with small number of neighbors or by considering that they are typically weak.

- Section 3.4.2: It is not clear how they arrived to the conclusion “The ST filter thus defines a backbone significantly different from the one obtained from either the DP or the ECM filters, even at fixed number of edges.”...which actually seems to contradict to their conclusion a couple of lines later “The various backbones seem thus to all retain similar sets of ties with large weights, while differing when assessing the significance of pairs of nodes with smaller numbers of interactions.”

- Section 3.4.4: The analysis on triangles was completed exclusively using ST edges, by saying that “the other filters do not use temporally resolved null models”. This is indeed true, but all methods, DP, ECM and SL, filter links and not events. Thus I think if they qualify for comparison of dyadic interactions, they equally (goodly or badly) qualify for the comparison of triadic motifs.

- In Fig.5: curves show some significant changes in the number of triangles at $\sim 10^{-5}$ and $\sim 10^{-13}$. Do the authors have any intuition about the reasons behind these changes?

Minor comments:

- Page 2 paragraph 3: The authors mention that their model can decide about a link with a single interaction whether it is significant or not. This is a strong claim and should be better clarified in the text.

- Data section: I would suggest to the authors to report some further details about the pre-treatment of the SocioPattern data, regarding de-noising, merging or not consecutive interactions, etc. Naturally, these details can be summarized in the Materials and Methods section.

- Table 1: It would be great to add a column indicating the number of links in the aggregated temporal networks. The number of contacts in some of the SocioPattern data looks very high regarding the length of observation.

- After Eq.2: "G is the cumulative distribution function...." add "between 0 and c"
- SI is used as an abbreviation of "Supplementary Information" in various places without introducing this abbreviation
- Section 3.4.2, 2nd and 3rd paragraph: I had difficulties to understand results explained here related to S5 and S6, especially that they are neither much explained in the SI.
- Fig.S3: what does the dashed line assign?

Reviewer #2 (Remarks to the Author):

The authors present a statistical method to filter out insignificant links in temporal networks by combining time windowing and the so-called fitness model. For a given significance level, their significant tie filter removes links between nodes that occur with high probability based on the product of the nodes' activity levels, which they determine with a maximum likelihood estimate based on the temporal data. With this approach, they can consider temporal motifs with more than two nodes and illustrate with triadic interaction. I have two major concerns:

First, while this method is the first to identify temporal backbones, and therefore should stimulate new work in the network science community, the presented results do not highlight the value of the information that the method can extract that other static network methods cannot extract. Summary statistics such as the number of identified significant ties (Fig 2) and overlap with some static methods (fig 3) or general statements such as the ones about triadic relationships provide little insight. These tests and comments give the impression that the authors have written the manuscript for a specialized journal. Results that combine specific, novel and interesting observations in a number of systems and a synthetic benchmark would be more convincing and accessible to a broad audience. The discussion about networks with community structure goes in this direction, but to some degree, they undermine the results when they write: "Note also that this temporal null model does not contain any a priori knowledge of the group structure of the nodes. An interesting generalization could be to superimpose group labels or node properties (e.g., gender or age for nodes representing individuals) and interaction probabilities depending on the nodes' properties." Moreover, reference 3, for example, argues for different models for different systems, which the authors must address.

Second, the method depends on time windowing, which ignores dynamics within the time windows: The approach introduces a time scale that can obscure the results. While the authors claim that the results do not depend on this time scale (Fig S4), I think these results say more about the summary statistics than the actual results. In the end, the level of aggregation seems arbitrary. Moreover, the authors acknowledge that the current manuscript provides little insight about the effect of the time scales in the final sentence: "Another direction of research would be to define a backbone at a finer resolution, namely that would be composed of (sets of) significant interactions instead of ties or set of ties."

In summary, before I can recommend publication in Nature Communications, the authors must more convincingly demonstrate the advantages over existing methods and why this generalization is better than other potential generalizations in a manuscript that targets a broader audience.

Reviewer #3 (Remarks to the Author):

The authors propose a method to extract the backbone of significant ties in temporal networks. The filter is based on a temporal null model that gives the probability that a pair of nodes interact in a time interval as a function of their activity levels, which are estimated from the empirical data using maximum likelihood. The authors compare the temporal filter with static filters on social face-to-face contact networks and an interbank financial network, and find that their method finds more significant edges and less biased towards large weights. The significant ties in aggregated networks with a clearcut community structure are mainly intra-community.

Filtering complex networks in a meaningful way is a difficult and relevant problem with no unique solution. The method proposed in this work seems sound and easy to implement and apply. The results presented in the manuscript look promising but are insufficient. Moreover, some technical aspects should be revised.

Even if the expression for the p-value can be easily derived from Eq. (2), it would be advisable to include its explicit expression in the manuscript.

There is a possible issue with Type I errors. The filter is not corrected for multiple comparisons and this could inflate the number of rejections of a true null hypothesis. This is important for each tie and could be even more important for the assessment of significant temporal motifs beyond single edges. The authors must evaluate the effects of applying controlling procedures. False discovery rate controlling procedures or even the more conservative family-wise error rate should be considered and their effects discussed.

The comparison of the temporal filtering method with static filtering methods in the aggregated network is interesting but biased. When using the ST filter the authors are evaluating the unexpected persistence of the interaction between a pair of nodes over time given their estimated activities, while applying a static filter on the aggregated gives a different information about anomalies in the distribution of the total number of interactions among the neighbors or in the distribution of interactions in the edges under a maximally random null model. In other words, one thing is how persistent is a tie and another is how persistence is distributed in the network. The authors should also compare with other alternatives that take into account the temporal nature of the system. One very simple possibility would be to apply the static filters to the networks in every time interval. Another possibility is to compare with other temporal filters.

Fig S5 is interesting and should be combined with Fig 3 in the main text. From Fig 3, it seems that when the number of preserved edges is similar, the filters greatly overlap, specially with DP, as soon as the number of edges is enough to provide minimal statistical quality (more than 200 edges in the backbones already give values of the Jaccard above 0.5). Is 80% similarity significantly different? For me, this is a strong signature of being strongly similar although not equivalent.

The weight distributions in Fig S6 are quite similar and somehow limited. The authors should look for other datasets with more heterogenous weight distributions to check more carefully their claims at the end of section 3.4.2.

In fact, the face-to-face social contact networks look very similar in all the analyzed features. How dependent are the results on the specific biases introduced by the data collection procedure for these kind of networks? It is highly recommended that the authors include in the study other datasets representing temporal networks in different domains, for which data is available.

The interpretation of inter-community edges as random interactions in section 3.4.4 is inaccurate. At the basis of the null model is that the interactions correspond to a stochastic process. I recommend to rephrase the discussion in terms of being significantly persistent in time or not. Overall, the finding about intra- and inter-community edges is not surprising.

It would be nice to see the total number of empirical snapshot edges as a function of the estimated activity values as compared to the expected number given by the null model.

Why has the interbank network been excluded from so many figures? Results for this network has only been shown in Figs S3 and S4, and in Fig 4c.

Relevant citations are missing, e. g. EPJ Data Science, December 2014, 3:27, Fast filtering and animation of large dynamic networks.

Response to the Referees' comments

First of all, we would like to thank the three anonymous reviewers for valuable comments and suggestions. We have substantially revised the manuscript and addressed each comment, as explained in detail below. We believe that the manuscript has strongly benefited from the reviewers' feedback and is now greatly improved. Additionally, in the revised version of the manuscript all substantial changes are **highlighted in red**.

Response to Reviewer #1's comments

Authors of this manuscript propose a method to identify significant links in temporal networks, this way constructing the backbone of the underlying structure. As the authors underline, there is a need for the development of such methods for the better analysis and reduction of recently available large temporal interaction datasets. In this sense the paper is timely and important. The manuscript is well written and the proposed methodology, as introduced, is scientifically sound.

Response: We thank the reviewer for these very positive comments.

However, I would like to ask the authors to address my various comments listed below. Some of the comments may be due to my misunderstanding, thus their clarification should be easy, while some other comments may highlight important conceptual points which needs to be addressed.

1. Major comments:

One of the cornerstone quantity of the proposed method is node activity. However, the authors miss to define what node activity means in case of real temporal networks. Is it the total number of observed interactions? Is it the number of interactions per time window? Activity appears as a parameter in Eq.1 and in the definition of the maximum likelihood estimator but it should be introduced earlier for better understanding.

Response: The activity of a node is in our case not directly observable: it is defined in the null model as an intrinsic variable of each node, and the activities of the nodes are estimated for each data set using the maximum likelihood estimation as described in Methods. It is thus a latent variable. Intuitively however, the activity of a node is expected to be related, as pointed out by the reviewer, to the number of its interactions. To check this, we show in Figure S1a the relationship between the estimated activity and the strength (i.e., the sum of all temporal edges involving from a node) for all nodes. We moreover show in Fig. S1b that the empirical strength (x-axis, 'data') is very close to the number of temporal edges for each node in the null model (i.e., $\tau \sum_{j \neq i} u(a_i, a_j)$, on the y-axis). We also

compare in Fig. S1c the total number of temporal edges in the data and in each corresponding null model.

We have added a section in the SI (S1) about the relationship between activity, strength and degree, and comments about the nature of the variable “activity” in the main text.

2. My main concern about the paper might be a confusion. The aim of the paper is to filter links from an already existing structure by deciding whether interactions on a link appear more frequently than one would assume solely from the activity levels of the two ending nodes. My first question is why the authors compute the null model between every pairs of nodes in the network? One does not need to filter non-existing links, on the other hand defining the likelihood this way may give the impression that a fully connected structure is considered in the definition.

Response: Thank you for raising an important point that needs to be clarified. The null model is not directly informed by the existence of links between pairs of nodes, only indirectly through the MLE procedure described in Methods, in which all pairs are considered, including the ones with $m_{ij}^o = 0$. The null model thus assumes that any pair of nodes could a priori interact (the probability $a_i a_j$ being always positive). This approach makes sense because the null model should capture a hypothetical situation in which the nodes would choose their neighbors at random, given their activities. This does not mean however that the actual realizations of the null model yield fully connected structures: actually, as shown in Fig S1, the number of temporal edges in the null model is equal to the one observed.

On the other hand, as noted by the reviewer, we do not need to filter non-existing links, so that we compute and compare the distribution under the null model (Eq 2) only for the set of existing (i.e., present in the data) links, in order to extract significant ties only from this set of edges. In other words, the comparison between the real network and a “randomized” network (the null model), in which the number of temporal edges of each node is preserved, allows us to detect pairs exhibiting overly many interactions that cannot be explained by random chance, given the nodes activities (namely, significant ties). To reinforce clarity, we have added comments below Eq. 1 and mention that we compute Eq 2 only for interacting pairs of nodes.

3. I understand that the method focuses on dyadic interactions in the computation of the backbone, while neglecting any aggregated network information, e.g., node degree. However, degree is a very important property in this setting as it determines the number of neighbors with whom a node needs to “share” its own activity. For example, significance of a link may be very different for an less active node with small degree than for

an less active node with high degree. Since the aim here is not to filter temporal networks “on the run”, but the dynamics of an already observed network, the aggregated degrees are already known in advance. This way considering degrees in the null model definition might be possible.

Response: We are grateful for this interesting suggestion. Incorporating information about the node degree in the estimation procedure could indeed enhance the accuracy of the proposed filtering method. However, on the one hand we would like the null model to include as few parameters as possible, and on the other hand, introducing this information in the null model and in the MLE procedure is neither straightforward nor doable in a unique manner. For instance, we could specify a matching function as $u = a_i a_j \zeta(K_i, K_j)$, but it means that we would add many additional parameters. Note also that, contrarily to the DP filter that determines whether a link is significant *for a node*, significance of a link is here (as in the ECM filter) determined globally.

Prompted by the reviewer’s comment, we have however considered this issue in more details: as the reviewer pointed out indeed, the lack of degree information could be a problem if the node degree carries essential information on the likelihood that the node constructs a significant tie. On the other hand, the activities estimated from the null model might already carry some information about the node degree, in which case additional information about the degree might not be essential.

To investigate this, we show in Fig. S3 that there exists a negative correlation between the fraction K^{sig}/K of significant ties of a node and its activity (K denotes the aggregate degree and K^{sig} the number of significant ties of a node). Nodes with lower activities (and thus smaller degrees, see Fig S2) have overall small probabilities in the null model to construct ties, and thus have higher chances that their ties turn out to be significant. This goes in the direction of the reviewer’s comment that node degree has some importance (“*significance of a link may be very different for an less active node with small degree than for an less active node with high degree*”).

Secondly however, we measure the correlation between aggregate degree and estimated activity: this is presented in Figure S2 and section S1.2 of the SI, and we find a strong positive correlation in all data sets. This indicates that the differences in node degrees are, at least partially, captured by the distribution of estimated activities.

Thirdly, we have computed, at each activity level (using bins of activity values to have enough data points), the correlations between degree K and the fraction K^{sig}/K of significant ties of a link. We find no significant correlations, which implies that the difference in degree is not informative once activity is controlled for.

To summarize, there exists a general tendency that the fraction of significant ties among all ties is higher for low-degree (low-activity) nodes than for high-degree (high-activity) nodes. However, if activity is controlled for,

the difference in the fraction of significant ties cannot be explained by the difference in degree, which implies that the activities already incorporate degree information.

4. Related to my previous comment, considering no degree information, inter-community links naturally appear mostly insignificant. Inter-community links are typically maintained by small number of interactions, however, they may be significant for a node with small number of neighbors or by considering that they are typically weak.

Response: The reviewer is right in that “(inter-community links) *may be significant for a node with small number of neighbors*”, because nodes with small number of neighbors generally have low activities (as illustrated in Fig. S2) and the nodes with smaller activities have higher chance of having significant ties, as discussed in the previous comments.

However, it turns out that the chance that an edge linked to a given node is significant is different between inter- and intra-community edges even at fixed node degree. We explore this indeed in the new Fig. S13, which shows for each node its fraction of inter-community and intra-community significant edges for $\alpha = 0.01$, vs. its degree: the fraction of inter-community edges identified as significant is generally lower than that of intra-community edges. Note that the figure shows that this property is generally true for *all* the filtering methods examined here and it is most apparent for the ST filter. Given this observation, the property that inter-community edges tend to be less significant than intra-community edges reflects an intrinsic difference between inter- and intra-community edges, and the ST filter allows us to exploit such a difference most efficiently.

5. Section 3.4.2: It is not clear how they arrived to the conclusion “The ST filter thus defines a backbone significantly different from the one obtained from either the DP or the ECM filters, even at fixed number of edges”... which actually seems to contradict to their conclusion a couple of lines later “The various backbones seem thus to all retain similar sets of ties with large weights, while differing when assessing the significance of pairs of nodes with smaller numbers of interactions.”

Response: Thank you for pointing out sentences that could confuse the reader. The two conclusions are actually compatible since the ties with large weights are much less numerous than the ties with small weights (the weight distributions are broad). So even if the sets of ties with large weights are similar, this concerns a relatively small number of ties, and the differences in the sets of ties with small weights can dominate when computing the Jaccard coefficient.

To avoid confusion, we have removed the sentence mentioned by the referee: “The ST filter thus defines a backbone significantly different from..”. Moreover, we now give the definition of the cosine similarity in the main

text and Fig. 4 now integrates the corresponding plots, in addition to the ones giving the Jaccard values.

6. Section 3.4.4: The analysis on triangles was completed exclusively using ST edges, by saying that the other filters do not use temporally resolved null models. This is indeed true, but all methods, DP, ECM and SL, filter links and not events. Thus I think if they qualify for comparison of dyadic interactions, they equally (goodly or badly) qualify for the comparison of triadic motifs.

Response: This is indeed an important point and we have now developed our explanations to make it clearer. As written by the reviewer, the examined filters are developed to extract significant links, rather than events, and do this based on null models examining the distribution of weights on these links. Several points are important here. First, our analysis is shown for triads as an illustration but, as described in 3.3, it is possible to assign a significance to general temporal structures. Second, the crucial point is that the significance concerns triads with *simultaneous* interactions. By definition of the static filters, which use only aggregate networks, it is not possible to include temporal information or constraints and thus to detect significant *simultaneous* triads or other temporal structures: even if a triad could be detected as significant from a static point of view, there is no guarantee that this triad would correspond to simultaneous interactions, it could instead correspond to any temporal order of the involved interactions and in particular could correspond to three nodes who never interact simultaneously. Most importantly, these static filters could not either detect any other general temporal structure. Third, from the definition of these filters, there is no clear extension allowing us to compute the significance of a triangle (or other static structures) other than the definition of a significant triangle as being composed of three significant edges: the corresponding result in figure 6b would then be trivial. To be more specific, the DP filter identifies edge significance by assuming that the weight distribution among the edges emanating from a focal node should be close to uniform if there is no preferential bias. It is designed to see the significance of bilateral relationships but not triadic relationships. The ECM filter compares the weight distributions among edges suggested by the null model (namely, the enhanced configuration model) and the data, so it is also not suitable to investigate the significance of triads.

Note that the significance of temporal structures could however be defined in the ECM-R, but once again only by defining a structure as significant iff all the events composing it are significant. Once again, no significant structure could then include non-significant dyads, contrarily to what we obtain in Fig. 6b.

We have added some comments in the section on triadic relationships to highlight these points, and we show in the SI (Fig. S15) the number of simultaneous triads for the ECM-R filter, where a significant triad is de-

defined as (i) the three corresponding edges exist simultaneously in at least one snapshot and (ii) they are all significant in at least one snapshot (the same for all three). The resulting number of significant triads decreases very fast as α decreases.

7. In Fig.5: curves show some significant changes in the number of triangles at $\sim 10^{-5}$ and $\sim 10^{-13}$. Do the authors have any intuition about the reasons behind these changes?

Response: We suspect that 10^{-15} is due to the limit of computer precision (which is 10^{-16}), and therefore any values close to the limit would be treated as almost zero in numerical computation. As for the other thresholds, $\sim 10^{-5}$, we do not really have any intuition about its value. Actually different datasets yield different shapes of the curve, and the drop is not exactly the same in all data sets, it starts at $\sim 10^{-5}$ in some cases and ends at this value in other cases.

8. Minor comments:

Page 2 paragraph 3: The authors mention that their model can decide about a link with a single interaction whether it is significant or not. This is a strong claim and should be better clarified in the text.

Response: Thank you for raising a point that needs to be explained more explicitly. In fact, the sentence “among all the pairs of nodes with at least one interaction in the data” means simply that we apply the filter to all ties that have interacted at least once, hence have an edge in the aggregated network: pairs of nodes with no interactions do not have an edge and hence are not considered. So, a priori any edge with at least one interaction could result significant. Note that this is the case also for other filters: for instance in the DP filter, if a node i has only one single interaction with only one other node, then its single link will be found significant.

Prompted by the comment of the reviewer, we have actually checked this issue. We have found significant dyads in the Workplace data with only one interaction over the whole data duration (at $\alpha = 0.01$). In the other data sets, we also found some significant pairs with only two or three interactions.

We have modified the sentence and added some comments on this in the text (in the discussion about the weight distributions).

9. Data section: I would suggest to the authors to report some further details about the pre-treatment of the SocioPattern data, regarding de-noising, merging or not consecutive interactions, etc. Naturally, these details can be summarized in the Materials and Methods section.

Response: We created a new section “Data processing” in Methods, in which we provide explanations about the data processing procedures.

Moreover, we have expanded the data description in the section “Data” in the main text.

10. Table 1: It would be great to add a column indicating the number of links in the aggregated temporal networks. The number of contacts in some of the SocioPattern data looks very high regarding the length of observation.

Response: Following the comment, we added a new column showing the number of aggregate edges in Table 1.

11. After Eq.2: “ G is the cumulative distribution function....” add “between 0 and c ”

Response: To make clear the definition of CDF G , we modified the explanation given below Eq. 2.

12. SI is used as an abbreviation of “Supplementary Information in various places without introducing this abbreviation

Response: We introduce the abbreviation “SI” when it first appears (i.e., in the second paragraph after Eq. 1).

13. Section 3.4.2, 2nd and 3rd paragraph: I had difficulties to understand results explained here related to S5 and S6, especially that they are neither much explained in the SI.

Response: We have modified the presentation of these results: we now give the definition of cosine similarity in the main text and show both the results of the Jaccard index and of the cosine similarity in the same figure so that the reader can capture the differences between filtered edges measured by the Jaccard index and by the cosine similarity.

14. Fig.S3: what does the dashed line assign?

Response: We have added an annotation, “ $M^* = M$ ”, for the dashed line in the figure (now Fig S1c).

Response to Reviewer #2's comments

1. First, while this method is the first to identify temporal backbones, and therefore should stimulate new work in the network science community, the presented results do not highlight the value of the information that the method can extract that other static network methods cannot extract. Summary statistics such as the number of identified significant ties (Fig 2) and overlap with some static methods (fig 3) or general statements such as the ones about triadic relationships provide little insight. These tests and comments give the impression that the authors have written the manuscript for a specialized journal. Results that combine specific, novel and interesting observations in a number of systems and a synthetic benchmark would be more convincing and accessible to a broad audience. The discussion about networks with community structure goes in this direction, but to some degree, they undermine the results when they write: "Note also that this temporal null model does not contain any a priori knowledge of the group structure of the nodes. An interesting generalization could be to superimpose group labels or node properties (e.g., gender or age for nodes representing individuals) and interaction probabilities depending on the nodes properties." Moreover, reference 3, for example, argues for different models for different systems, which the authors must address.

Response: We thank the reviewer for his/her comments that our work is likely to "stimulate new work": this is precisely the reason why we think it is suited for a journal with a broad audience, as it might be of interest to many readers. We acknowledge that the previous version of the manuscript was considering only a limited set of data sets and was lacking validation on a synthetic benchmark.

To improve on this and follow the reviewer's suggestion, we have brought two major additions to the paper. First, we have added the following three new data sets (all publicly available) representing temporal networks of different systems other than face-to-face human interactions: (i) a temporal network of emails exchanged between members of a European research institution ("Email"), (ii) the data for trips between bike sharing stations taken by customers of London Bicycle Sharing Scheme ("LondonBike") and (iii) the domestic airline network in the UK ("UK-airline"). This means that we have now tested our methods in a wide variety of real-world systems: social (Highschool, Primaryschool, Workplace, Hospital), economic (Interbank), online (Email), human mobility ("LondonBike") and infrastructure systems (UK-airline).

Moreover, we have added a new section of validation in which we consider as well a synthetic benchmark: in this model, nodes are endowed with an intrinsic "activity" and interact at each time step with probability given by the product of their activities, exactly as in the temporal null model;

in addition to this background of random interactions, we select a known fraction of “strong” ties for which the number of interactions is increased with respect to the null model. It is then interesting to investigate whether the various filters can identify these stronger ties as “significant” for certain significance levels. We show that the ST filter indeed allows us to extract these strong ties more efficiently than other methods do. In particular, we find that the DP, ECM and ECM-R (newly added) filters fail to detect these strong ties as significant as soon as $\alpha < 0.01$, while the ST filter recovers a large fraction of these ties over a very broad range of significance values. The fact that the ST filter does not classify all the strong links as significant (and yields almost no false positive) leads us also to argue that the fraction of significant ties in the ST filter should be regarded as a conservative measure of the true fraction of significant ties. Overall, the numerical results validate the ST filter on the benchmark and suggest that filtering methods based on aggregate information are not suitable as a way to extract significant ties in a temporal setting even though their presence have a non-negligible influence on the weight distribution of the aggregate network.

Regarding the issue of networks with community structure, our point is that the null model should incorporate as little information as possible to be as general as possible and to limit the number of parameters. The generality of the model allows us to use it in a wide variety of cases, as now shown by the addition of several data sets. Having a general model also means that the model can be more easily modified, extended or refined, which is what we meant by our sentence about a possible addition of group labels or node properties. Different extensions might then indeed be more adapted to different systems, but it is first important to assess the results of this baseline model, which is what we do here; moreover, in many cases no group information is available in the data, and/or no community structure is present, and in such cases the best one can do is indeed to use such an agnostic null model.

2. Second, the method depends on time windowing, which ignores dynamics within the time windows: The approach introduces a time scale that can obscure the results. While the authors claim that the results do not depend on this time scale (Fig S4), I think these results say more about the summary statistics than the actual results. In the end, the level of aggregation seems arbitrary. Moreover, the authors acknowledge that the current manuscript provides little insight about the effect of the time scales in the final sentence: “Another direction of research would be to define a backbone at a finer resolution, namely that would be composed of (sets of) significant interactions instead of ties or set of ties.”

Response: This is indeed an interesting point and we hope that our work will stimulate further inquiries in this direction. It is correct that our method introduces a time scale on which the data is aggregated. This is in

fact the case in most investigations of temporal structures (e.g., temporal motifs) and automatically detecting relevant time scales in data is a highly non-trivial task. In our case, this time scale can be changed and one can examine how the results change when Δ changes. We acknowledge that the previous version of the manuscript did not include a detailed enough analysis on the effect of Δ , and we have now investigated this issue more thoroughly: we have added a new figure (Fig. S12) to quantify the overlap between the sets of significant pairs detected at different resolutions. The figure shows that increasing the value of Δ (i.e., decreasing the temporal resolution) slightly reduces the number of significant pairs for a given significance level, which lowers the Jaccard index (which remains however rather high even for very different values of Δ). Moreover, Fig S12c shows the overlap coefficient between the sets of significant ties at different resolutions: this coefficient is given by $S(I_{ST}^{\Delta}, I_{ST}^{\Delta'}) \equiv |I_{ST}^{\Delta} \cap I_{ST}^{\Delta'}| / \min(|I_{ST}^{\Delta}|, |I_{ST}^{\Delta'}|)$ and its values remain very high (> 0.9 , much higher than the Jaccard coefficient). This shows that the set of significant ties at a lower resolution Δ' is almost included in the set of significant ties at higher resolution Δ . Therefore, we can conclude that changing the temporal resolution Δ slightly affects the number of significant pairs, but that the significant pairs detected at a lower Δ contain the ones detected at larger values of Δ .

3. In summary, before I can recommend publication in Nature Communications, the authors must more convincingly demonstrate the advantages over existing methods and why this generalization is better than other potential generalizations in a manuscript that targets a broader audience.

Response: As written by the reviewer, ours is actually the first method to identify temporal backbones. We are not aware of any other generalization, that would not be a simple filtering snapshot by snapshot, and in fact we now also consider such a simple generalization (ECM-R, in which the ECM is applied in each snapshot). Our manuscript thus presents the method and presents a comparison mostly with static filters. Moreover, we have now added a synthetic benchmark and three new data sets corresponding to a broad range of systems.

To summarize, the main interests of the filter we present are:

- i) it can extract significant ties more efficiently than other methods do, as the analysis of synthetic networks indicate. Notably, the possibility of type-II errors (false negative) is much lower than that under different methods;
- ii) it can efficiently identify intra-community edges even for a fixed number of edges;
- ii) most importantly, thanks to the underlying temporal null model, it can go beyond the identification of significant ties and can identify significant temporal structures (with an arbitrary significance) such as triads of *simultaneous* interactions, while static filters cannot impose temporal

constraints nor be easily generalized to structures (other than by trivially imposing significance of all ties involved in the structure).

Finally, it can be modified, refined or extended and our work is likely to stimulate other works in this direction, which justifies its publication in a journal with a broad audience.

Response to Reviewer #3's comments

The authors propose a method to extract the backbone of significant ties in temporal networks. The filter is based on a temporal null model that gives the probability that a pair of nodes interact in a time interval as a function of their activity levels, which are estimated from the empirical data using maximum likelihood. The authors compare the temporal filter with static filters on social face-to-face contact networks and an interbank financial network, and find that their method finds more significant edges and less biased towards large weights. The significant ties in aggregated networks with a clearcut community structure are mainly intra-community.

Filtering complex networks in a meaningful way is a difficult and relevant problem with no unique solution. The method proposed in this work seems sound and easy to implement and apply. The results presented in the manuscript look promising but are insufficient. Moreover, some technical aspects should be revised.

Response: We thank the Reviewer for this positive appreciation of our work. We describe below (and in the answers to the other Reviewers' comments) how we have improved our manuscript, adding new material and tests on additional data sets as well as on synthetic networks.

1. Even if the expression for the p-value can be easily derived from Eq. (2), it would be advisable to include its explicit expression in the manuscript.

Response: We added a formal definition of the p-value below Eq. 2.

2. There is a possible issue with Type I errors. The filter is not corrected for multiple comparisons and this could inflate the number of rejections of a true null hypothesis. This is important for each tie and could be even more important for the assessment of significant temporal motifs beyond single edges. The authors must evaluate the effects of applying controlling procedures. False discovery rate controlling procedures or even the more conservative family-wise error rate should be considered and their effects discussed.

Response: Thank you for the suggestion. Following this comment, we applied the most conservative correction method, namely Bonferroni correction, in showing the fraction and the number of significant ties (Figs. 2

and 3). Note that the application of this correction corresponds in fact to a rescaling of the significance level, as the filters we consider are tuned by this parameter: as a result, the curve of the number of significant ties vs. significance level (Figures 2, 3 and S7) is shifted to the right with respect to the curve without correction (as the x-axis showing α is in in log-scale). Note also that the numerical investigations performed with the synthetic network suggest that the fraction of significant ties detected by the ST filter would itself be a conservative measure (see answer to point 1 of Reviewer 2); there is always a possibility that a fraction of true significant ties may not be detected as long as the activities of end nodes are sufficiently high. Interestingly, in the synthetic data there are instead virtually no type-I errors (namely, false positive), as seen in Fig. S5.

3. The comparison of the temporal filtering method with static filtering methods in the aggregated network is interesting but biased. When using the ST filter the authors are evaluating the unexpected persistence of the interaction between a pair of nodes over time given their estimated activities, while applying a static filter on the aggregated gives a different information about anomalies in the distribution of the total number of interactions among the neighbors or in the distribution of interactions in the edges under a maximally random null model. In other words, one thing is how persistent is a tie and another is how persistence is distributed in the network. The authors should also compare with other alternatives that take into account the temporal nature of the system. One very simple possibility would be to apply the static filters to the networks in every time interval. Another possibility is to compare with other temporal filters.

Response: We agree with the reviewer: our filter is qualitatively not comparable with static filters because the definition of “edge significance” is different in a fundamental way that the referee correctly pointed out. In fact, we present the results for static filters in order to see how *different* our temporal filter is from the existing static filters (rather than if our filter is “*better*”): our comparison was mostly aimed at showing the interest of exploiting the temporality of the data and thus a temporal null model instead of using only static aggregated networks and static null models. We are grateful to the reviewer for the suggestion to add a comparison with a baseline method using temporality. We have adopted this suggestion and build a temporal filtering method by implementing the ECM filter in each temporal snapshot; we then regard node pairs as being connected by significant ties if they are identified as significant in at least one snapshot. This method is called “ECM-R” (i.e., ECM-repetitive) throughout the paper. To apply ECM-R, we need to have a sequence of weighted networks, each of which is created by aggregating temporal edges over a time interval. In all data sets except UK-airline, weights are given by the number of interactions in a time interval. For the UK-airline data set, weights are defined by the number of passengers since the data for the number of

interactions (i.e., flights) in a year is not available.

4. Fig S5 is interesting and should be combined with Fig 3 in the main text. From Fig 3, it seems that when the number of preserved edges is similar, the filters greatly overlap, specially with DP, as soon a the number of edges is enough to provide minimal statistical quality (more than 200 edges in the backbones already give values of the Jaccard above 0.5). Is 80% similarity significantly different? For me, this is a strong signature of being strongly similar although not equivalent.

Response: We thank the reviewer for the suggestion that we have implemented. In the new version of the figure (Fig. 4), we now present both the Jaccard index and the cosine similarity for the Highschool data. We also added panels for the newly introduced algorithm, ECM-R. The corresponding figures for the other data sets are presented in Figs. S8 and S9. We also acknowledge that our presentation of the results could be confusing and we agree that the sets of edges identified by the various methods are similar but not equivalent. As pointed out by the reviewer, there are some cases in which Jaccard index is above 80%, but these cases in fact correspond to rather low significance levels (e.g, $\alpha > 0.1$). This was not clear in the original version, so in the caption of Fig. 4 we added a comment about the range of significance levels we considered: $\alpha = [10^{-17}, 0.5]$. Moreover, to avoid confusion, we have removed the sentence starting from “The ST filter thus defines a backbone significantly different from..”. In addition, as mentioned above, results concerning the cosine similarity are now presented in the main text.

5. The weight distributions in Fig S6 are quite similar and somehow limited. The authors should look for other datasets with more heterogenous weight distributions to check more carefully their claims at the end of section 3.4.2.

Response: As discussed above, we have added the analysis of three new data sets corresponding to very different systems, and updated the figures of the weight distributions (Fig. S10). In this figure, to eliminate the influence of differences in the number of edges, we fix the number of filtered edges to calculate weight distributions. We have also updated the discussion about the weight distributions: except for the DP filter, which tends to retain larger weights, the distributions of weights obtained by all filters are broad and detect significant ties at all scales.

6. In fact, the face-to-face social contact networks look very similar in all the analyzed features. How dependent are the results on the specific biases introduced by the data collection procedure for these kind of networks?

It is highly recommended that the authors include in the study other datasets representing temporal networks in different domains, for which data is available.

Response: As discussed also in the answer to Reviewer 2, we have now added analysis of three very different data sets collected by different methods and representing different systems: the Email network of a European research institution (“Email”), the data for trips between stations taken by customers of the London Bicycle Sharing Scheme (“LondonBike”), and the UK domestic airline network (“UK-airline”). The descriptions of the data sets are provided in the Data section. We believe that the introduction of these data sets has increased the generality of our results.

7. The interpretation of inter-community edges as random interactions in section 3.4.4 is inaccurate. At the basis of the null model is that the interactions correspond to a stochastic process. I recommend to rephrase the discussion in terms of being significantly persistent in time or not. Overall, the finding about intra- and inter-community edges is not surprising.

Response: Thank you for pointing out this, as our formulation was somehow too strong. We have rephrased several sentences to make this statement more precise and explain that the inter-community edges can be explained by the null hypothesis.

8. It would be nice to see the total number of empirical snapshot edges as a function of the estimated activity values as compared to the expected number given by the null model.

Response: Thank you for the suggestion. This comment is related to the comment 1 of Reviewer #1. Figure S1 shows on the one hand the strength of nodes (i.e., their total number of temporal edges) vs. their estimated activity, and on the other hand that the total number of empirical snapshot edges is successfully predicted by the null model.

9. Why has the interbank network been excluded from so many figures? Results for this network has only been shown in Figs S3 and S4, and in Fig 4c.

Response: Thank you for the suggestion. While the figures in the main text display the results for a subset of data sets, we put the other figures for different data sets in Supplementary Information (SI).

10. Relevant citations are missing, e. g. EPJ Data Science, December 2014, 3:27, Fast filtering and animation of large dynamic networks.

Response: Thank you for this reference. We have added a citation in the Introduction.

Reviewers' comments:

Reviewer #1 (Remarks to the Author):

I carefully read the resubmitted manuscript entitled as "The structured backbone of temporal social ties" by Kobayashi, Takaguchi, and Barrat. The authors addressed all my comments with the expected resolution. Below I list a few typos and editorial comments to help the authors to further improve their manuscript.

- a discussion on the limitations of the proposed methods is missing from the manuscript.

- the authors mention "the null model" in the abstract while it is not clear what it is at this point for the reader

- Page 10, the 1st sentence of the paragraph starts as "We validate our filtering method" is too long.

- In the same paragraph the authors write: "Interestingly, at a given level of significance, our method generally identifies more significant edges than other filters and, as other filters based on null models, is able to detect significant edges at all scales of interaction intensity." This would give the impression that their model is not based on null models, which is not true.

- Page 6, last line: "decreases. and" -> "decreases and"

- Page 7 last line: "type-I errors" - not defined or referenced

- Page 9, in the last paragraph the 1st sentence is too long.

- Page 14 1st line: "timw" -> "time"

Reviewer #2 (Remarks to the Author):

The authors have satisfactorily responded to my technical comments. However, to recommend publication in Nature Communications, I have two remaining issues:

The first is about significance and impact. The authors "find that (i) at given level of statistical significance, our method identifies more significant ties than methods considering temporally aggregated networks, and (ii) when a community structure is present, most significant ties are intra-community edges, suggesting that the weights of inter-community edges can be explained by the null model." But as a reader and potential user, this isn't convincing. How do I know that identifying more ties is better? The synthetic networks are a chance to contrast the new method with current methods. What would I miss with current methods? What would I gain with the new method? How will this change my understanding? Of the synthetic benchmarks? Of the real examples? Presenting the first significant tie filter for temporal networks is not enough. The author must quantitatively and qualitatively demonstrate the gain and significance of the results. Also, "A more detailed comparison shows that the difference does not come from ties with large numbers of interactions, which are similarly selected by all methods, but rather by the fact that our filter uncovers more significant pairs of nodes with small number of interactions." in the Discussion is just a difference and not a clear gain.

The second is about clarity. Many compound sentences are hard to follow because they don't tell a single story from opening to resolution about one thing. For example, the following one starts with filtering non-essential links and ends with the availability of time-resolved networks: "Filtering them [non-essential links] out and extracting a set of relevant connections, the "network backbone", is a non-trivial task, and methods put forward until now do not address time-resolved networks, whose availability has strongly increased in recent years." It is a big leap from start to end, which requires intermediate steps in separate sentences.

Reviewer #3 (Remarks to the Author):

The manuscript has been improved and the results in the revised version seem more clear and robust now. However, I cannot yet recommend the paper for publication.

There is one important issue that remains to be addressed, which refers to the relationship between strength and degree in the considered networks. This is in fact an important point since the results

presented in this work could depend on this relation. It is also important for clarifying how the proposed filter compares with other alternatives, in particular the DP filter.

In page 9 and the caption of Fig. S10, it is claimed that the DP filter is almost equivalent to a simple thresholding of the weights. On the one hand, the claim seems unsupported by the results shown in Fig. S10. The reported curves for ST and DP significant edges seem actually very similar to the naked eye, a direct comparison with a global threshold filter would be necessary to support the claim.

On the other hand, and more importantly, it is already known that the DP filter and the global threshold strategy give similar results when applied to a complex network where weights are uncorrelated with the topology, whenever their probability distribution has a well defined average, see Supporting Information of Ref. (13). In the uncorrelated situation, the strengths and the degrees of the nodes are related in a trivial way and topology and weights are decoupled.

If this is the case for the real datasets considered in this work, as suggested by the claims of the authors, then there is no need to use the DP filter for the analyzed datasets, and the authors should have restricted to use a simpler global threshold instead of the DP filter.

In any case, even if the relations between strengths and activities and degrees and activities are shown in Fig. S1, the authors need to check and report the direct relationship between strength and degree in the analyzed datasets. If a trivial relationship is found, the authors need to include in the manuscript the analysis of networks where the weights are coupled with the topology in a non-trivial way, using both synthetic and real data. If a non-trivial relationship, like a power-law strength-degree correlation, is found for some dataset but the DP filter still gives similar results to a global thresholding procedure, then this would be an indication of some problem with the implementation of the DP filter.

Minor details:

The analysis of the DP filter applied to each temporal snapshot should be also included, analogously to the ECM-R method.

The citation of Gemmetto et al. (17) in section S3.1 is incorrect. The first time that it was noted that the significance of an i - j edge in the DP filter is not equivalent to that between j and i even in undirected networks was in Ref. (13).

Response to the Reviewers' comments

We are grateful to the Reviewers, who have carefully read our answers and revised manuscript, and recognize that we have satisfactorily answered their comments. As the Reviewers have raised some additional comments, we provide below a detailed answer to each. We have moreover revised our manuscript, adding some discussion and making long sentences clearer, and in the revised version of the manuscript all substantial additions are **highlighted in red**.

Response to Reviewer #1's comments

I carefully read the resubmitted manuscript entitled as "The structured backbone of temporal social ties by Kobayashi, Takaguchi, and Barrat. The authors addressed all my comments with the expected resolution. Below I list a few typos and editorial comments to help the authors to further improve their manuscript.

Response: We thank the reviewer for these very positive comments and his/her careful reading of our manuscript. We have added a discussion of some limitations of our method, and also corrected the typos, shortened sentences throughout the text, and corrected the sentence that could be misinterpreted as stating that we do not use null models.

Response to Reviewer #2's comments

The authors have satisfactorily responded to my technical comments. However, to recommend publication in Nature Communications, I have two remaining issues:

1. *The first is about significance and impact. The authors “find that (i) at given level of statistical significance, our method identifies more significant ties than methods considering temporally aggregated networks, and (ii) when a community structure is present, most significant ties are intra-community edges, suggesting that the weights of inter-community edges can be explained by the null model.” But as a reader and potential user, this isn't convincing. How do I know that identifying more ties is better? The synthetic networks are a chance to contrast the new method with current methods. What would I miss with current methods? What would I gain with the new method? How will this change my understanding? Of the synthetic benchmarks? Of the real examples? Presenting the first significant tie filter for temporal networks is not enough. The author must quantitatively and qualitatively demonstrate the gain and significance of the results. Also, “A more detailed comparison shows that the difference does not come from ties with large numbers of interactions, which are similarly selected by all methods, but rather by the fact that our filter uncovers more significant pairs of nodes with small number of interactions.” in the Discussion is just a difference and not a clear gain.*

Response: We thank the Reviewer for this comment. We had tried in our manuscript to make clear the interest of our method, but the Reviewer's comment shows that we needed to put more emphasis on its importance. We have thus extended the Discussion section to emphasize more the advantages of our method. Our main points are the following: first, the fact that our null model takes into account temporality is already an important novelty, as previous filters consider only the aggregated networks. So we claim that the first impact of our work indeed consists in the novelty of being able to deal with temporal networks, which have become the object of a lot of attention and studies.

Second, we agree with the Reviewer that it is important to say why identifying more ties is better. There are actually several points here:

- the investigation of Jaccard coefficient and cosine similarity between the backbones obtained by our filter and the other filters shows that they all tend to retain similar sets of ties with large weights. This means that the difference between filters is mainly due to ties with small weights, and therefore that our filter identifies more ties with small weights. These important ties with small weight are indeed more difficult to identify, so it is an advantage of our filter that it manages to identify more of them;

- we show in the synthetic data set that our filter identifies correctly more ties *known a priori to be significant* than the other filters; in such a case, it is clear that “more is better”, as there are (almost) no false positives and we can compare the results to a ground truth;
- in the synthetic data, we have verified that our filter indeed tends to identify better than the other filters the ties with small numbers of interactions (new figure S6); note that we have added in the SI (section S4.2) an additional synthetic data set with two types of reinforced ties: the first type is identified by our filter, while the other type is built to be identified by the other filters but not by our filter, as it corresponds to anomalously reinforced weights that do not correspond to an anomalous temporal activity. Our numerical simulations confirm that our significant ties filter detects correctly the temporal significant ties, while the other filters detect the statically reinforced weights;
- the set of significant ties identified by our filter is very stable on a broad range of significance values (and much more stable than for other filters). This makes it possible to define a backbone in a more robust way, without fine-tuning the significance level, and, in the synthetic case, to retrieve also in a more robust way the set of significant ties.

Third, we show that our filter, thanks to its temporal nature, allows us to go beyond the identification of significant ties, and can also identify significant *temporally constrained structures*. Importantly, these significant structures are defined without making reference to the individual ties that compose them, and indeed do not necessarily coincide with sets of significant ties. Such an endeavour is by definition impossible with static filters. For the ECM-R or DP-R, which take into account some temporality, it is possible to define significant structures, but only in a trivial and non-principled way, by defining them as sets of significant ties: it is then impossible to identify significant structures that would include non-significant ties. The ability to identify such temporally constrained significant structures is thus a very clear advantage of our method, stemming directly from the use of a temporal null model.

Fourth, the stability of the set of significant ties on a broad range of significance values also paves the way to the definition of a representation of the temporal network as the superposition of a backbone and the temporal null model. Such a representation could not be obtained from static filters based on static null models.

Finally, our method can be extended (we show in the SI an example with time-varying activity) and form the basis for other filtering methods for temporal networks. We have made these points more explicit in our revised version.

We have made all these points clearer in the new version of the manuscript.

2. *The second is about clarity. Many compound sentences are hard to follow because they don't tell a single story from opening to resolution about one thing. For example, the following one starts with filtering non-essential links and ends with the availability of time-resolved networks: "Filtering them [non-essential links] out and extracting a set of relevant connections, the "network backbone, is a non-trivial task, and methods put forward until now do not address time-resolved networks, whose availability has strongly increased in recent years." It is a big leap from start to end, which requires intermediate steps in separate sentences.*

Response: We have streamlined the text and carefully shortened long sentences in order to improve clarity.

Response to Reviewer #3's comments

The manuscript has been improved and the results in the revised version seem more clear and robust now. However, I cannot yet recommend the paper for publication.

Response: We thank the Reviewer for acknowledging that we have improved our manuscript and answered her/his comments.

There is one important issue that remains to be addressed, which refers to the relationship between strength and degree in the considered networks. This is in fact an important point since the results presented in this work could depend on this relation. It is also important for clarifying how the proposed filter compares with other alternatives, in particular the DP filter.

In page 9 and the caption of Fig. S10, it is claimed that the DP filter is almost equivalent to a simple thresholding of the weights. On the one hand, the claim seems unsupported by the results shown in Fig. S10. The reported curves for ST and DP significant edges seem actually very similar to the naked eye, a direct comparison with a global threshold filter would be necessary to support the claim.

On the other hand, and more importantly, it is already known that the DP filter and the global threshold strategy give similar results when applied to a complex network where weights are uncorrelated with the topology, whenever their probability distribution has a well defined average, see Supporting Information of Ref. (13). In the uncorrelated situation, the strengths and the degrees of the nodes are related in a trivial way and topology and weights are decoupled.

If this is the case for the real datasets considered in this work, as suggested by the claims of the authors, then there is no need to use the DP filter for the

analyzed datasets, and the authors should have restricted to use a simpler global threshold instead of the DP filter.

In any case, even if the relations between strengths and activities and degrees and activities are shown in Fig. S1, the authors need to check and report the direct relationship between strength and degree in the analyzed datasets. If a trivial relationship is found, the authors need to include in the manuscript the analysis of networks where the weights are coupled with the topology in a non-trivial way, using both synthetic and real data. If a non-trivial relationship, like a power-law strength-degree correlation, is found for some data set but the DP filter still gives similar results to a global thresholding procedure, then this would be an indication of some problem with the implementation of the DP filter.

Response: We emphasize that the Reviewer’s comment does not concern the filtering method we propose but rather the DP filter and its comparison with a global thresholding, i.e., both static filtering methods. Several points can be commented here:

- as the DP filter is more fundamentally grounded than a pure global thresholding, and is a well-known filtering method, it seems natural to consider it as one of the reference static filters, rather than the global thresholding;
- in p.9, we were writing that both filters were “almost equivalent”. In the caption of the previous figure S10 (figure S13 in the revised manuscript), we wrote instead that the DP filter was “closer to” a global thresholding than the other filters. The former claim was excessive and we apologize for this lack of clarity. We show in the figure below that indeed by using the Kullback-Leibler (KL) divergence between the weights distributions, that the latter statement is more correct: the DP is not equivalent to a global thresholding, but closer to it than other filters. Note that it is also clear from the new figure S6 that DP is not equivalent to a global thresholding. We have therefore updated the text and also mention that the similarity to a global thresholding filter can be quantified;
- the figure below also shows the scatterplots of strength vs degree. Large fluctuations are found in most data sets. We also note that the KL divergence between DP and global thresholding depends a lot on the specific data set;
- the comparison between DP and global thresholding is actually only a sidenote and does not impact our results: these are two static filters, and our point is neither to compare two static filters to each other nor to criticize any, but to introduce a novel filter that takes into account the temporal nature of the data. The important result is that the distributions of the weights of the significant ties is broad, showing that the filters find significant ties at multiple scales. We therefore argue that this point needs not to be developed further in our manuscript.

Minor details:

Figure R1: a) Strength vs. degree in the various data sets. (b) Kullback-Leibler divergence between the weight distributions obtained with the various filters and with a global thresholding. The divergence is consistently smaller for the DP filter.

The analysis of the DP filter applied to each temporal snapshot should be also included, analogously to the ECM-R method.

Response: We have added this analysis, labeled DP-R, in several figures.

The citation of Gemmetto et al. (17) in section S3.1 is incorrect. The first time that it was noted that the significance of an i - j edge in the DP filter is not equivalent to that between j and i even in undirected networks was in Ref. (13).

Response: We have corrected this reference.

REVIEWERS' COMMENTS:

Reviewer #2 (Remarks to the Author):

The authors have satisfactorily responded to my remaining comments and I recommend publication.

Reviewer #3 (Remarks to the Author):

My previous comments were clearly not in the direction of asking for a comparison between the DP filter and a global thresholding procedure, something known that was done and published in the past as I was pointing out in my previous report. My comment was indicating an inconsistency in the previous version of their work, where they claimed that the DP filter is almost equivalent to a simple thresholding of the weights, a claim that was unsupported by the results that were shown in Fig. S10, and that would have undermined the use of DP in front of a global threshold if true. The new version of the manuscript has been corrected and is now more precise.